# ACCORD: Autoregressive Constraint-satisfying Generation for Combinatorial Optimization with Routing and Dynamic Attention

## Abstract

Large Language Models (LLMs) have demonstrated impressive reasoning capabilities, yet their direct application to NP-hard combinatorial problems (CPs) remains underexplored. In this work, we systematically investigate the reasoning abilities of LLMs on a variety of NP-hard combinatorial optimization tasks and introduce **ACCORD**: **A**utoregressive **C**onstraint-satisfying generation for **CO**mbinatorial optimization with **R**outing and **D**ynamic attention. ACCORD features a novel dataset representation and model architecture that leverage the autoregressive nature of LLMs to dynamically enforce feasibility constraints, coupled with attention-based routing to activate problem-specific LoRA modules. We also present the ACCORD-90k supervised dataset, covering six NP-hard combinatorial problems: TSP, VRP, Knapsack, FlowShop, JSSP, and BinPacking. Extensive experiments demonstrate that our ACCORD model, built on an 8B-parameter Llama backbone, consistently outperforms standard prompting and input-output methods, even when compared to much larger LLMs, such as gpt-4. Ablation studies further show that our output structure enhances solution feasibility. To the best of our knowledge, this is the first large-scale, end-to-end framework for exploring the applications of LLMs to a broad spectrum of combinatorial optimization problems.

## 1 Introduction

Large Language Models (LLMs) have rapidly established themselves as versatile engines for reasoning across a broad spectrum of tasks, encompassing arithmetic, commonsense logic , Thoppilan et al. (2022), Chowdhery et al. (2023), Brown et al. (2020). Among the prominent strategies enabling such capabilities is the Chain-of-Thought approach, which allows these models to decompose complex problems into sequential, interpretable steps Wei et al. (2022b).

Recent efforts have sought to adapt these reasoning techniques to address more advanced optimization tasks. Combinatorial optimization problems (CPs) are decision-making challenges where the goal is to select an optimal arrangement or subset from a large, discrete set of possibilities. Classic examples include the Traveling Salesman Problem (TSP), Vehicle Routing Problem (VRP), and Job Shop Scheduling Problem (JSSP), which have widespread applications in logistics, manufacturing, and artificial intelligence Lenstra et al. (1979). Due to their NP-hard nature, even moderately sized instances possess a combinatorial explosion of potential solutions, rendering brute-force approaches infeasible. As a result, practical methods typically rely on heuristics or approximation algorithms to provide near-optimal solutions within reasonable time frames. As NP-hard problems, CPs present huge obstacles in practical settings Oroojlooyjadid et al. (2020). Presently, the predominant paradigm in industry relies on metaheuristic algorithms—sophisticated combinations of simple, efficient heuristics—for solving CPs under various constraints. However, the success of these heuristics is often highly sensitive to the specific structure and requirements of each problem, necessitating tailored approaches for optimal results.

At the same time, investigations into leveraging LLMs for combinatorial problem solving have revealed significant research gaps. While the latest breakthroughs highlight the promise of LLMs in

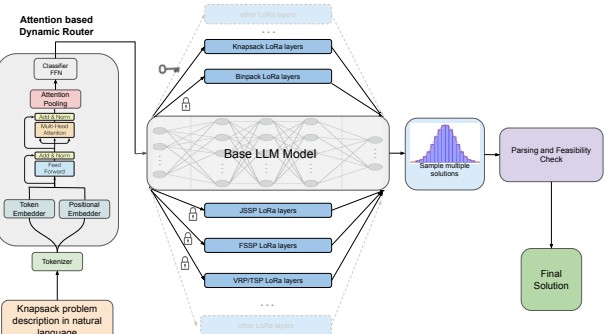

Figure 1: Overview of the ACCORD inference pipeline. As an example, a knapsack problem described in natural language is provided as input, then Attention based Dynamic router5 activates the corresponding LoRA layer specialized for knapsack tasks. Multiple candidate solutions are generated via sampling, each checked for feasibility. The best feasible solution is returned as the final output. Note that the pipeline generalizes to other combinatorial problems in the same manner; knapsack is shown here for illustration.

diverse reasoning scenarios Abgaryan et al. (2024), Iklassov et al. (2024), Wei et al. (2022a), Zhang et al. (2022), their full potential in the context of combinatorial optimization remains largely untapped. Applying LLMs directly to these problems presents unique challenges: LLMs are trained primarily for natural language generation, not for enforcing strict combinatorial constraints, leading to issues such as hallucinations (plausible but infeasible solutions) Huang et al. (2022), lack of optimality, and limited interpretability Valmeekam et al. (2022). Furthermore, the absence of systematic search or explicit constraint mechanisms means LLM outputs can violate feasibility or fail to improve upon prior attempts. Recent advances have begun to explore the application of large language models (LLMs) to combinatorial optimization (CO). Numerous prompting-based approaches have been tested on CO tasks Yang et al. (2023); Huang et al. (2024); Mittal et al. (2024); Wei et al. (2022b); Zhou et al. (2022); Madaan et al. (2023); Iklassov et al. (2024), demonstrating progress in solution quality and constraint handling. However, to date, there has been no comprehensive study evaluating a unified fine-tuned LLM-based framework for NP-hard CO problems across multiple domains.

In this work, we address this gap by introducing **ACCORD** (**A**utoregressive **C**onstraint-satisfying generation for **CO**mbinatorial optimization with **R**outing and **D**ynamic attention), a novel framework for testing the reasoning capabilities of LLMs on combinatorial optimization problems. Our main contributions are as follows: (i) We propose the ACCORD90k supervised dataset for TSP, VRP, Knapsack, FlowShop, JSSP, and BinPacking, employing an ACCORD representation that explicitly encodes problem constraints by leveraging the autoregressive nature of LLMs;(ii) we develop a model architecture that leverages attention-based dynamic routing and specialized Low-Rank Adaptation (LoRA) modules for different CO tasks; (iii) extensive ablation studies demonstrate that our method achieves lower optimality gaps and higher solution feasibility than both the traditional list-of-lists representation and state-of-the-art prompting techniques (including GPT-4 with the Code Interpreter enabled). Notably, it achieves substantial improvement percentage difference in feasibility over the list-of-list representation, with gains of 24.86% in FlowShop, 7% in JSSP, 4% in Knapsack, and 2% in BinPacking, 10% in VRP and TSP, problems. To the best of our knowledge, this is the first work to demonstrate large-scale, end-to-end combinatorial problem solving with LLMs, offering new directions for testing symbolic reasoning and optimization within language models.

## 2 RELATED WORK

### 2.1 HEURISTIC AND MACHINE LEARNING APPROACHES ON CO PROBLEMS

Combinatorial optimization has been tackled with both heuristic and exact methods. Simple priority dispatching rules (PDRs), such as shortest processing time or earliest due date, are computationally efficient but often yield suboptimal solutions due to their greedy nature Lenstra et al. (1979). Meta-heuristics (e.g., simulated annealing, tabu search, genetic algorithms) offer improved solution quality,

and exact approaches like the shifting bottleneck procedure Adams et al. (1988), mixed-integer programming, and constraint programming can find optimal solutions for small instances, though at high computational cost Roy & Sussmann (1964); Goel et al. (1996). Recently, machine learning, particularly deep reinforcement learning (RL) and graph neural networks (GNNs) have advanced combinatorial optimization Zhang et al. (2020a); Khalil et al. (2017); Kool et al. (2019). RL methods treat scheduling as sequential decision making, learning dispatching policies via environment interaction Zhang et al. (2020a). GNNs encode jobs and machines as nodes, enabling permutation-invariant representations and, when combined with RL, can model complex dependencies Khalil et al. (2017). Attention-based and sequence-to-sequence models further enhance performance on tasks like TSP and VRP, often utilizing iterative refinement Kool et al. (2019).While prior work explores prompting, search, and task-specific fine-tuning for CO, we focus on a unified, multi-task feasibility-aware *representation* paired with dynamic adapter routing.

## 2.2 Large Language Models in Combinatorial Optimization

The advent of LLMs has introduced new paradigms for CO. Early work explored whether LLMs could generate solutions through prompting Yang et al. (2023), Huang et al. (2024), Mittal et al. (2024), Wei et al. (2022b) Zhou et al. (2022), Madaan et al. (2023), Iklassov et al. (2024). Prompting-based strategies, such as OPRO, involve iterative refinement based on feedback, while methods for VRP employ self-debugging and verification to enhance feasibility Huang et al. (2024). However, scalability remains a challenge, as even strong prompting techniques struggle on larger or more complex instances Mittal et al. (2024). Recent research has explored a variety of prompting strategies to leverage LLMs for solving combinatorial optimization (CO) problems. The **Input-Output (IO)** method presents the LLM with multiple examples of input and corresponding output solution pairs. The LLM is then prompted to generate an output solution in the same format as the provided examples. This approach relies on the LLM's ability to generalize the mapping from input to output based on observed patterns. In **Chain-of-Thought (CoT)** prompting, the LLM is guided to produce a sequence of intermediate reasoning steps, or "thoughts," before arriving at the final answer Wei et al. (2022b). This technique encourages the model to break down complex CO tasks into structured, stepwise reasoning, improving both transparency and solution quality. **Least-to-Most (LtM)** prompting strategy aims to decompose a complex problem into a sequence of simpler subproblems, solving them incrementally Zhou et al. (2022). Each subproblem builds upon the solutions of previous ones, enabling the LLM to tackle challenging CO tasks through a series of manageable steps. **Self-Refinement (SR)** is an iterative prompting technique wherein the LLM first generates an initial solution, then provides feedback on its own output, and finally refines the solution based on this feedback Madaan et al. (2023). The process repeats until a satisfactory solution is reached. **Self-Guiding Exploration for Combinatorial Problems (SGE)** autonomously generates multiple thought trajectories for a given CO task Iklassov et al. (2024). Each trajectory represents a distinct heuristic approach, inspired by metaheuristics. SGE decomposes these trajectories into actionable subtasks, executes them sequentially, and refines the results to ensure optimal solutions. Prompting examples for each type of instance can be found in Appendix B.1. Fine-tuning LLMs for CO tasks is another active area Abgaryan et al. (2024),Masoud et al. (2024) . Abgaryan et al. (2024) showed that fine-tuned LLM on job-shop scheduling, demonstrates significant improvements in solution quality. Similarly, Masoud et al. (2024) applied fine-tuning to TSP instances with promising but size-limited results. Hybrid methods integrate LLMs into evolutionary or search frameworks, where the LLM guides genetic operations or receives feedback from constraint solvers to iteratively improve solutions Liu et al. (2023); Wan et al. (2024); Awasthi et al. (2025). While promising, these approaches often entail significant computational overhead and still face scaling hurdles.

## 3 Preliminaries: Overview of Classic Combinatorial Optimization Problems

Combinatorial optimization involves searching for the best solution from a finite set of possibilities. Formally, given a set of feasible solutions $\mathcal{S}$ and an objective function $f : \mathcal{S} \to \mathbb{R}$, the goal is to find

$$s^* = \arg\min_{s \in \mathcal{S}} f(s)$$

or, in some cases, to maximize $f(s)$ depending on the problem. Details of each of the formal definition of each combinatorial optimization task can be found in Appendix A

Table 1: Optimality gap (%) comparison for prompting methods using GPT-4 with Code Interpreter (IO, CoT, SR, LtM, SGE) vs. ACCORD (Llama 8B) on five combinatorial optimization tasks (Knapsack, Bin Packing, TSP, VRP, JSSP). Gap is defined as $(\mathrm{obj} - \mathrm{opt})/\mathrm{opt} \times 100\%$; N/A indicates no feasible solution. Sizes 5, 8, 12 denote increasing instance scales per task (comparable down a column, not across tasks). For each (task, size) combination, 50 instances were used, shared across all methods. For every instance–method pair, a fixed budget of 60 candidate solutions was generated(under identical sampling setting, temperature=1.0, top_p=1, top_k=50), and the best one, based on the lowest gap was selected. Bold values (with an asterisk) mark the best mean gap at each size. Structured prompting (LtM, SGE) narrows gaps relative to IO/CoT/SR, while ACCORD method with small Llama 8B model backbone achieves consistently low gaps with high feasibility.

| Size | Method | Knapsack | BinPack | TSP | VRP | JSSP |
|---|---|---|---|---|---|---|
| | IO (GPT-4) | 90.1 | 108.2 | 100.3 | 102.0 | 105.3 |
| | CoT (GPT-4) | 66.9 | 78.2 | 81.2 | 78.2 | 79.4 |
| | SR (GPT-4) | 62.0 | 77.4 | 71.6 | 72.5 | 71.7 |
| 5 | LtM (GPT-4) | 21.6 | 40.0 | 43.6 | 40.7 | 44.1 |
| | SGE (GPT-4) | 8.1 | 9.1 | 8.3 | 11.9 | 9.3 |
| | IO (Llama 8B) | N/A | N/A | N/A | N/A | N/A |
| | ACCORD (Llama 8B) | 3.9* | 0.0* | 0.6* | 1.0* | 0.0* |
| | IO (GPT-4) | 103.5 | 112.8 | 116.9 | 116.3 | 108.2 |
| | CoT (GPT-4) | 73.8 | 85.1 | 89.0 | 89.5 | 85.2 |
| | SR (GPT-4) | 72.6 | 86.3 | 85.6 | 83.3 | 78.4 |
| 8 | LtM (GPT-4) | 26.4 | 52.7 | 53.5 | 54.4 | 49.8 |
| | SGE (GPT-4) | 14.9 | 21.0 | 15.2 | 19.7 | 21.3 |
| | IO (Llama 8B) | N/A | N/A | N/A | N/A | N/A |
| | ACCORD (Llama 8B) | 7.4* | 0.0* | 1.8* | 1.0* | 5.0* |
| | IO (GPT-4) | 101.5 | 120.7 | 121.6 | 118.5 | 117.6 |
| | CoT (GPT-4) | 79.3 | 93.8 | 86.8 | 90.1 | 89.3 |
| | SR (GPT-4) | 77.1 | 82.2 | 88.6 | 88.4 | 87.0 |
| 12 | LtM (GPT-4) | 35.8 | 55.4 | 57.5 | 59.2 | 56.0 |
| | SGE (GPT-4) | 16.8 | 22.4 | 16.1 | 24.0 | 22.9 |
| | IO (Llama 8B) | N/A | N/A | N/A | N/A | N/A |
| | ACCORD (Llama 8B) | 5.1* | 2.6* | 2.9* | 2.2* | 12.4* |

# 4 MAIN METHOD: ACCORD REPRESENTATION FOR FEASIBILITY-AWARE SOLUTION GENERATION

A core challenge in applying Large Language Models (LLMs) to combinatorial optimization is the effective encoding of feasibility constraints within the generated solutions. Conventional representations, such as the "list of lists" format, provide direct encodings of solution sets, which are familiar to LLMs due to their prevalence in general-purpose data and code corpora(more details of the format are available in Appendix A.1). However, these representations are static, constraints are only checked after solution generation, offering limited guidance for incremental feasibility during the autoregressive decoding process. To address this limitation, we decided to utilize the autoregressive nature of the LLMs and developed a representation, which is specifically designed to leverage the autoregressive generation paradigm of LLMs. Unlike the list-based format, our representation decomposes solutions into a sequence of state transitions, with each step not only specifying the next element of the solution but also explicitly updating and exposing the relevant feasibility metrics (e.g., cumulative weights, distances, machine usage, or value). This design allows the model to compute and check constraints dynamically as each token is generated, closely mimicking the typical reasoning and verification process of a human solver.

**Formal ACCORD representation.** Consider a CO problem with decision sequence $X = (x_1, \ldots, x_T)$, constraints $\mathcal{C}(X)$, and objective $f(X)$. An ACCORD serialization is a sequence

$$S = (s_1, \ldots, s_T), \qquad s_t = (a_t, \Delta_t, u_t),$$

where $a_t$ is the action at step $t$, $\Delta_t$ is the incremental update to feasibility state variables, and $u_t = g(u_{t-1}, a_t)$ is the updated state summary that explicitly encodes (and textually asserts) constraint satisfaction. Generation factorizes autoregressively:

$$P(S) = \prod_{t=1}^{T} P\big(s_t \mid s_{<t}\big),$$

and terminates with a special token $\langle \mathrm{END} \rangle$, after which a verifier checks $u_T \models \mathcal{C}$.

**Knapsack (capacity $W$):**

$$s_t = \big(\texttt{item\_id}, (\Delta v, \Delta w), (v_{t-1}+\Delta v,\ w_{t-1}+\Delta w \leq W)\big).$$

**TSP/VRP:**

$$s_t = \big(\texttt{next\_node}, \Delta d = \text{dist}(n_{t-1}, n_t),\ D_t = D_{t-1}+\Delta d\big).$$

**JSSP:**

$$s_t = \big(\texttt{op}(j,k), \Delta t = p_{j,k}, \texttt{timeline}(M_{j,k}) \text{ updated without overlap}\big).$$

ACCORD representation embeds constraint satisfaction directly into the generation process. For instance, in the Knapsack problem, each item addition is accompanied by an explicit update of the running total value and weight, immediately verifying the capacity constraint at each step:

```
[[item_id, weight, value] -> value:  prev_v + value =
new_v, weight:  prev_w + weight = new_w <= capacity],
...
```

Similarly, for Bin Packing, the incremental assignment of items to bins is annotated with cumulative weights, ensuring that no bin exceeds its capacity as the sequence unfolds. Routing problems (VRP, TSP) and scheduling problems (JSSP) are analogously handled by tracking cumulative distances or machine times within the autoregressive output stream. Example of each of problem type in ACCORD representation is available in the Appendix A.1. This approach transforms the constraint satisfaction problem into a stepwise process, where feasibility checks are interleaved with generation. As a result, the LLM is naturally guided away from infeasible sequences, as each decision is immediately contextualized by the current state of the solution.

## 4.1 DATASET GENERATION

We generated synthetic supervised datasets for several CO problems using Google OR-Tools Google (2025) as the solver. For each instance, solutions were produced in both the conventional "list of lists" and ACCORD representations. Roughly 15,000 instances were created per task, using a compute node with 64 CPUs (Intel® Xeon® Gold 5218 @ 2.30GHz, 16 cores per socket, 2 threads per core, x86_64 architecture, 46-bit physical / 48-bit virtual addressing, 44MiB L3 cache).

**TSP & VRP:** Instances varied in location count ($N \in \{5, 8, \ldots, 100\}$) and number of vehicles ($V \in \{1, \ldots, 10\}$), with random coordinates and demands. OR-Tools solved these using the 'PATH_CHEAPEST_ARC' strategy. **Knapsack:** Item counts ($N \in \{5, \ldots, 100\}$) and difficulty were varied, influencing item properties and constraints. Optimal solutions were computed using OR-Tools, discarding instances that timed out. **Bin Packing:** Instances varied by item count, weight limits, and target bin numbers. Bin capacities were set accordingly, and OR-Tools was used to minimize bin usage under a timeout. **JSSP:** Job Shop Scheduling instances ranged from $10 \times 10$ to $100 \times 20$ jobs and machines, with random operation sequences and durations. Solutions minimized makespan using the CP-SAT solver. **FSSP:** Permutation Flowshop instances ranged from $5 \times 1$ to $50 \times 2$ and $2 \times 50$, with random processing times. Solutions were generated with the NEH heuristic Nawaz et al. (1983). For fine-tuning, a dataset of 15,000 instances was used per task, with 600 out of training distribution examples reserved for validation.

## 5 ROUTER ARCHITECTURE

To dynamically activate the appropriate LoRA layers for each combinatorial optimization problem, we use an attention-based Dynamic Router TextClassifier that selects the relevant LoRA weights based on the instruction text (see Figure 1). Our model builds on a transformer architecture enhanced to capture problem-specific features. Each input token $x_i$ is embedded with positional information and normalized as $\mathbf{E}' = \text{Dropout}(\text{LayerNorm}(\mathbf{E}_{\text{token}}(\mathbf{x}) + \mathbf{E}_{\text{pos}}(\mathbf{p})))$.

The resulting embeddings are projected to the hidden dimension and passed through several transformer layers with alternating multi-head attention and feed-forward sublayers, each followed by layer normalization. Token representations from the final transformer layer are pooled using attention-based pooling: $\mathbf{r} = \sum_{i=1}^{n} a_i \mathbf{h}_i$, and passed through a classification head defined by $\mathbf{y} = \mathbf{W}_2 \cdot \text{LayerNorm}(\text{GELU}(\mathbf{W}_1 \mathbf{r} + \mathbf{b}_1)) + \mathbf{b}_2$.

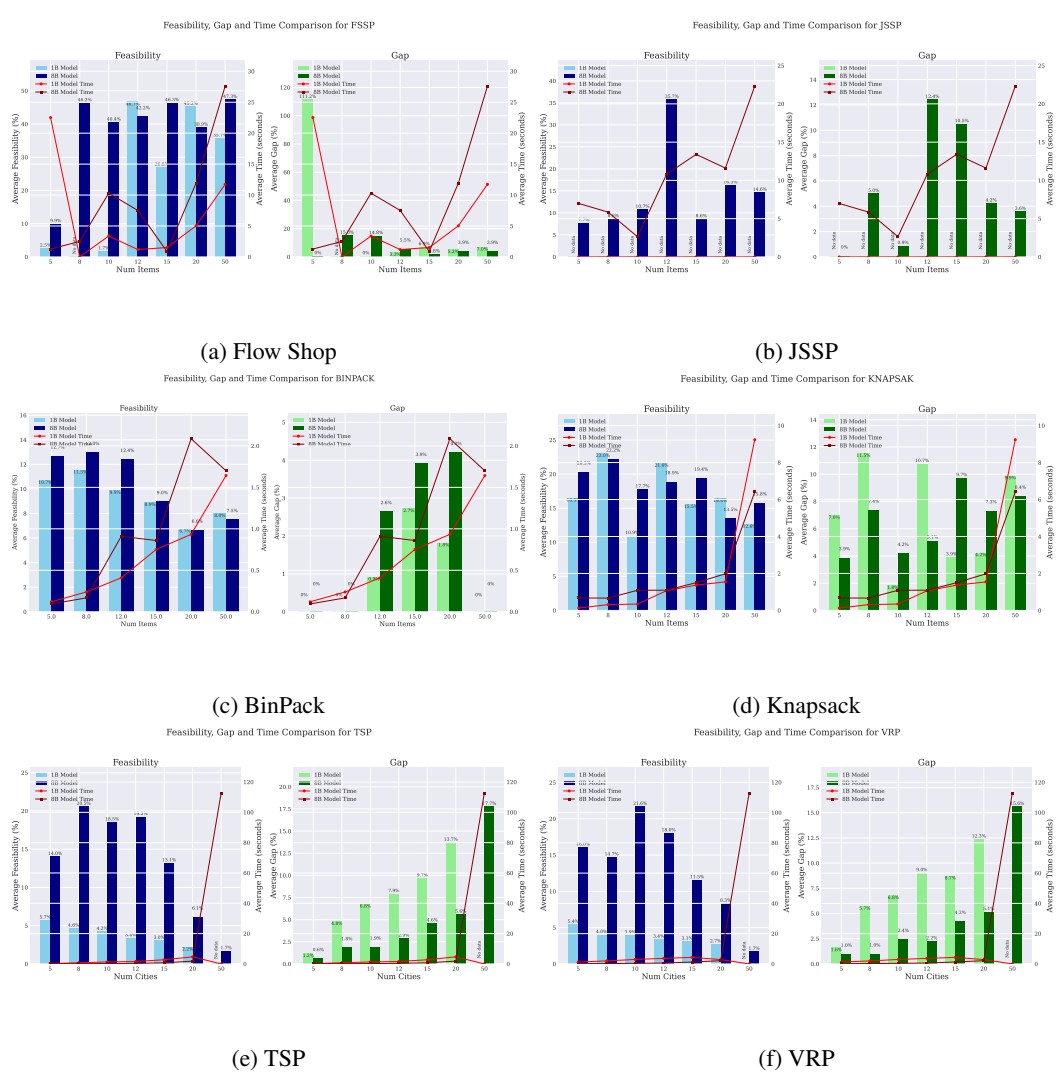

(a) Flow Shop

(b) JSSP

(c) BinPack

(d) Knapsack

(e) TSP

(f) VRP

Figure 2: This figure illustrates the performance of the Llama 3.1 (8B) and Llama 3.2 (1B) models in terms of the average gap percentage compared to the OR-Tools solution, where a lower gap indicates better performance. The left y-axis represents the average gap percentage, while the right y-axis corresponds to the running time in seconds. Bar plots indicate the average gap. The line plots depict the average running time per instance size, with the x-axis showing the problem size in terms of the number of nodes in the graph representation. Instances labeled as "No Data" indicate that, within a sampling budget of 60, the model failed to generate any feasible solution.

The pooled vector **r** enables instruction-based selection of problem-specific LoRA adapters via the predicted logits **y**. The router was trained using 1,000 example instances sampled from each combinatorial optimization task, resulting in a total training set of 6,000 instances. The data was split into 80% for training, 15% for validation, and 5% for testing. Further training details of the router network are provided in Appendix B. The impact of the router is analyzed in the ablation study presented in Appendix B.2.

# 6  TRAINING DETAILS

We conducted supervised fine-tuning using input-output pairs for two models from Meta: Llama 3.1 8B and Llama 3.2 1B. To minimize memory usage during training, we employed 4-bit quantized versions of these models and trained each for 2 epochs. For a fair comparison, we fine-tuned

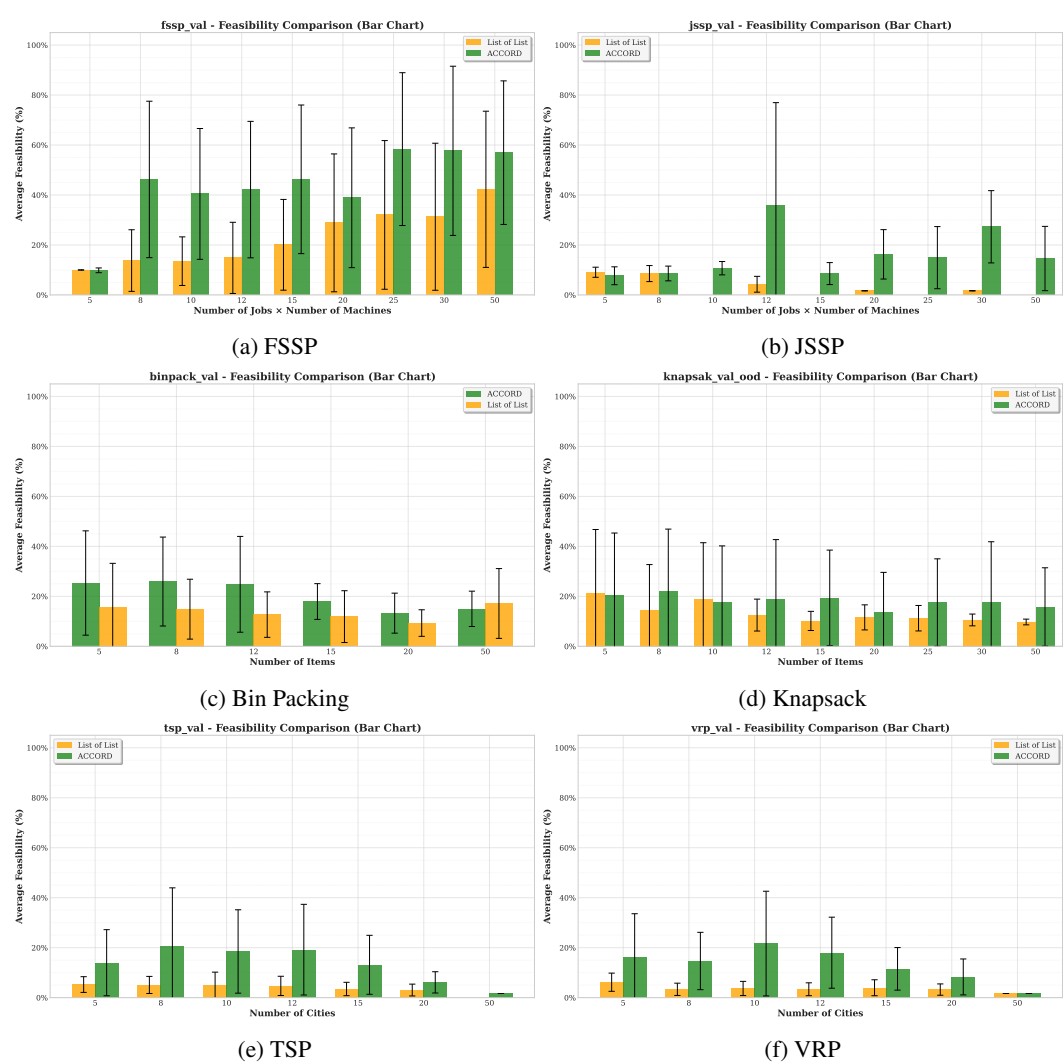

Figure 3: Average feasibility comparison with OR-Tools solution across different problem instance sizes; the higher the feasibility percentage, the better.

each model with the same hyperparameters, varying only the output representation: once using the list-of-lists format and once using the ACCORD format, while keeping the input and all other hyperparameters identical. We used Rank-Stabilized Low-Rank Adaptation (RSLoRA) Kalajdzievski (2023) with a rank of $r = 64$ and $\alpha = 64$. We fine tuned for 2 epochs, which required roughly 40 hours and about 30GB of GPU memory on Nvidia RTX A6000 GPU. We limited the context length of the model to 40k instead of the original 128k, to reduce memory consumption and increase the speed of fine-tuning. "Context length" refers to the maximum number of tokens (words or subwords) the model can process at once as input.

## 6.1 EMPIRICAL COMPARISON WITH LIST-OF-LIST REPRESENTATION

We empirically evaluate the impact of problem representation by fine-tuning Llama 3.1 8B on both list-of-list and ACCORD formats with identical hyperparameters and input (see Section 4.1), using a validation set of 100 out-of-distribution instances for each problem size ($n \in \{5, 8, 10, 12, 15, 20, 25, 30, 50\}$). The inference pipeline (Fig. 1) employs an Attention-Based Dynamic Router (Section 5) to select the appropriate LoRA branch, generating 60 candidate solutions per instance. Each solution is checked for feasibility, and the best feasible solution, i.e., the one with the lowest optimality gap is selected as the final output.

**Direction-aware optimality gap.** Let $J_{\text{model}}$ be the objective value of the model's best feasible solution and $J_{\text{opt}}$ the baseline/optimal objective. Set $\gamma = +1$ for minimization tasks (TSP, VRP, JSSP, FSSP, Bin Packing) and $\gamma = -1$ for maximization tasks (Knapsack).

$$\text{Gap}(\%) = 100\,\gamma\,\frac{J_{\text{model}} - J_{\text{opt}}}{|J_{\text{opt}}|}. \tag{1}$$

This yields $\text{Gap} \geq 0$ when the model is no better than the baseline and $\text{Gap} = 0$ when it matches it. We additionally report feasibility rate (%) as the fraction of generated candidates that satisfy all constraints. Our results (Fig. 3) show that, although list-of-list representation is familiar to LLMs, models trained with this format tend to ignore feasibility constraints, resulting in lower feasibility rates and higher optimality gaps. In contrast, the ACCORD representation explicitly encodes feasibility into the output, enabling the LLM to produce a larger proportion of valid and near-optimal solutions, particularly as the problem size increases. Table 1 further compares our method against various prompting strategies (see Section 2 for baselines) on both Llama 8B and GPT-4 with code interpreter enabled. Notably, while GPT-4 can potentially generate and execute solver code, our ACCORDbased method enables the LLM to generate solutions end-to-end without code execution. Inference using the SGE method Iklassov et al. (2024) strictly follows the procedure described in the original paper. Prompt examples for SGE can be found in Appendix Section B.1. For both our approach and all prompting baselines, 60 samples per instance are generated, and the best result is selected. ACCORD consistently outperforms prompting strategies across all 6 combinatorial optimization tasks, and achieves optimal solutions on smaller instances. We also assess the impact of model size on average gap, feasibility, and inference time (Fig. 2). The 8B model mostly outperforms the 1B model in feasibility and optimality gap, with only a moderate increase in inference time. For harder instances, such as JSSP, the 1B model fails to find feasible solutions within the sampling limit. Our results demonstrate that scaling from 1B to 8B parameters yields a significant 31.5% relative improvement in solution quality, reducing the average gap from 6.54% to 4.48%

The most substantial improvements were observed in routing problems, with TSP and VRP showing 65% and 54% relative gap reductions, respectively. The results on ACCORD on TSPLib with strong neural and heuristic baselines can be found in Appendix Table 11 and additional comparisons on randomly OOD generated TSP instances can be found in Table 10. Bin packing problems showed minimal sensitivity to model scale, with only a 1% improvement. In addition to our synthetic OR-Tools instances, we also evaluated ACCORD8B on Taillard permutation flow-shop benchmarks (50 jobs × 10 machines and 50 jobs × 20 machines; avg. gap $\approx 13.7\%$) and on job-shop benchmarks TAITaillard (1993) (15 × 15 to 50 × 20; avg. gap $\approx 21.7\%$) and DMUDemirkol et al. (1998) (20 × 15 to 50 × 15; avg. gap $\approx 22.1\%$) against standard heuristics (MWR/MOR/SPT) and the L2D neural scheduler (see Table 2 and 9).

Table 2: Comparison of different methods on the JSSP **TAI** benchmark (sampling budget = 60). Lower values indicate solutions closer to the optimal, representing better scheduling performance. An asterisk (*) denotes the best result based on the Percentage Gap. Classic JSSP heuristics (FDD/WKR, MOPNR, MWKR, SPT), whose gap values are unaffected by sampling and therefore do not include standard deviation, are described in Appendix B.2. Neural methods include *L2D*, *RASCLB*, and *ACCORD-Ours*.

| Method | 15x15 | 20x15 | 20x20 | 30x15 | 30x20 | 50x15 | 50x20 | Average |
|---|---|---|---|---|---|---|---|---|
| FDD/WKR | 47.45 | 50.57 | 47.57 | 45.01 | 56.30 | 37.72 | 42.80 | 46.77 |
| MOPNR | 44.98 | 47.97 | 43.68 | 45.59 | 48.23 | 31.25 | 39.24 | 42.99 |
| MWKR | 56.74 | 60.65 | 55.60 | 52.61 | 63.93 | 41.90 | 55.62 | 55.29 |
| SPT | 54.64 | 65.24 | 64.11 | 61.61 | 66.03 | 51.37 | 61.00 | 60.57 |
| L2D | 25.95 ± 3.37 | 30.03 ± 3.90 | 31.60 ± 4.11 | 33.02 ± 4.29 | 33.62 ± 4.37 | 26.15 ± 3.40 | 26.40 ± 3.43 | 29.54 ± 3.84 |
| RASCLB | 20.59 ± 2.47 | 25.31 ± 3.04 | 25.47 ± 3.06 | 27.27 ± 3.27 | 30.40 ± 3.65 | 20.69 ± 2.48 | 26.40 ± 3.17 | 25.16 ± 3.02 |
| ACCORD-Ours | **19.34 ± 1.93*** | **18.00 ± 1.80*** | **21.11 ± 2.11*** | **21.44 ± 2.14*** | **30.05 ± 3.00*** | **17.57 ± 1.76*** | **24.32 ± 2.43*** | **21.69 ± 2.17*** |

## 6.2 ABLATION STUDY ON LATENT SPACE PROXIMITY AND SOLUTION FEASIBILITY

To investigate the connection between latent representations and solution feasibility, we analyzed 500 TSP instances processed using both ACCORD and list-of-list formats. For each instance, we extracted hidden-state representations from the final transformer layer of the Llama 3.1 8B model with PCA dimensionality reduction, then computed the Euclidean distance between paired representations

from each format. We subsequently evaluated the feasibility of the solutions generated by both models. Statistical analysis revealed a significant negative correlation between latent distance and solution feasibility ($r = -0.1182$, $p = 0.0155$, $p < 0.05$), indicating that solutions whose latent representations are closer to those produced by the ACCORD format are more likely to satisfy constraints. This trend was further supported by quartile analysis, which showed feasibility rates consistently decreasing as latent distance increased. Notably, this relationship holds despite a large performance gap between the formats (71.4% feasible solutions for ACCORD vs. 1.6% for list-of-lists). These findings suggest that LLMs encode constraint satisfaction geometrically: solutions closer to the ACCORD manifold in latent space are more likely to be feasible. Thus, latent proximity can predict solution quality, indicating that neural solvers capture structural information about combinatorial constraints beyond explicit training signals.

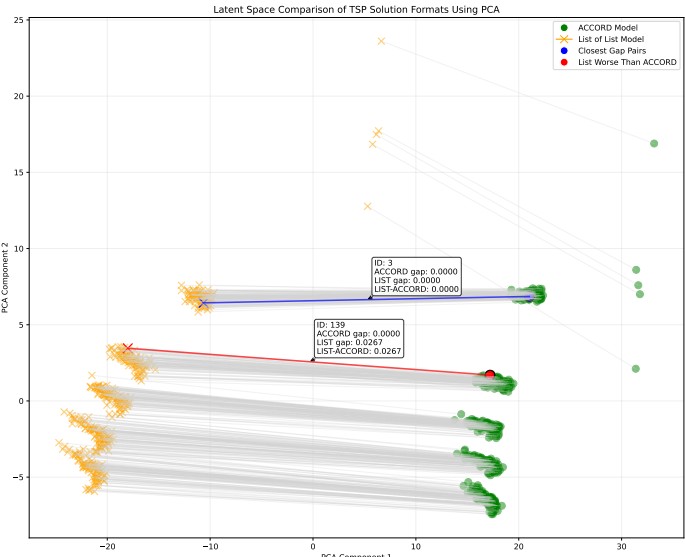

Figure 4: Latent representation distance versus solution feasibility on TSP problems, demonstrating negative correlation between distance and constraint satisfaction.

## 7 CONCLUSION

We introduced ACCORD a framework that encodes combinatorial constraints into an autoregressive text format and uses dynamic LoRA routing to probe an LLM's end-to-end ability on NP-hard tasks. On six standard benchmarks (TSP, VRP, FlowShop, JSSP, Knapsack, BinPacking), an 8B-parameter model trained with ACCORD achieves strong feasibility rates and competitive optimality gaps compared to prompting and a naïve list-of-lists format. ACCORD does not replace specialized solvers; rather, it probes how far small LLMs can go as feasibility-aware generators under a unified representation. We hope the dataset, grammar, and verifier lower the barrier to hybrid methods that blend neural generation with classical search.

## 8 LIMITATIONS AND FUTURE WORK

Despite its strong performance, ACCORD is bounded by the LLM's context window (limiting very large instances) and relies on LoRA adapters on an 8B-parameter model. In future work, we will investigate larger backbones (with full fine-tuning), expand the effective context via external memory or hierarchical encoding, and apply ACCORD to real-world, large-scale optimization scenarios.

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

# A APPENDIX

The code is publicly available at `https://github.com/starjob42/ACCORD` and the dataset at `https://huggingface.co/datasets/mideavalwisard/ACCORD`.

## FORMAL DEFINITION OF COMBINATORIAL OPTIMIZATION PROBLEMS

**Traveling Salesman Problem (TSP)** Given a list of cities and the distances between each pair of cities, what is the shortest possible route that visits each city exactly once and returns to the starting point. Mathematically, for $n$ cities $V = \{1, 2, \ldots, n\}$ and a distance matrix $D \in \mathbb{R}^{n \times n}$, we seek a tour (a permutation $\pi$ of all cities) that minimizes the total travel distance, where $\pi(n+1) = \pi(1)$ to ensure the tour closes:

$$\min_{\pi \in \mathcal{P}_n} \sum_{i=1}^{n} D_{\pi(i), \pi(i+1)}$$

**Vehicle Routing Problem (VRP)** The VRP extends the TSP to multiple vehicles. Given a depot, $n$ customers (with demands $q_i$), and a fleet of vehicles each with capacity $Q$, the goal is to design routes—each starting and ending at the depot—so that every customer is visited exactly once, no vehicle exceeds its capacity, and the total travel distance is minimized:

$$\min \sum_{k=1}^{m} \sum_{j=0}^{\ell_k} D_{v_j^k, v_{j+1}^k}$$

subject to

$$\bigcup_{k=1}^{m} \{v_1^k, \ldots, v_{\ell_k}^k\} = V \quad \text{(All customers served)}$$

$$\sum_{j=1}^{\ell_k} q_{v_j^k} \leq Q \quad \forall k \quad \text{(Capacity constraint)}$$

**Job Shop Scheduling Problem (JSSP)** JSSP schedules $n$ jobs, each as a sequence of operations on specific machines. Each operation $O_{j,k}$ requires machine $M_{j,k}$ for $p_{j,k}$ time units, following job order. Let $S_{j,k}$ and $C_{j,k}$ be the start and completion times. The objective is to minimize makespan:

$$\min C_{\max} = \max_j C_{j, \ell_j}$$

subject to:

$$\text{(Precedence)} \quad S_{j,k+1} \geq C_{j,k}$$
$$\text{(No machine conflicts)} \quad S_{j,k} \geq C_{j',k'} \text{ or } S_{j',k'} \geq C_{j,k},$$
$$\forall (j,k) \neq (j',k') \text{ with } M_{j,k} = M_{j',k'}$$

**Knapsack Problem (KP)** Given a set of items, each with a value and weight, what is the most valuable combination of items you can carry without exceeding the weight limit of your knapsack. With $n$ items (weights $w_i$, values $v_i$) and capacity $W$, choose $x_i \in \{0, 1\}$ (item picked or not) to solve:

$$\max \sum_{i=1}^{n} v_i x_i \qquad \text{s.t.} \qquad \sum_{i=1}^{n} w_i x_i \leq W$$

**Bin Packing Problem (BPP)**
Given a set of items of varying sizes, how can you pack them into the fewest number of fixed-size bins. For $n$ items of sizes $s_i \in (0, 1]$, assign them to bins of capacity $1$ so as to minimize the total number of bins $K$:

$$\min K$$

subject to:

$$\sum_{i \in B_k} s_i \leq 1; \quad \bigcup_k B_k = \{1, \ldots, n\}; \quad B_k \cap B_{k'} = \emptyset$$

for all $k$, $k' \neq k$. where $B_k$ is the set of items in bin $k$.

**Flow Shop Scheduling Problem (FSSP)**

Given $n$ jobs and $m$ machines, each job $j$ has processing times $p_{j,k}$ on machine $k$. Find a job sequence $\pi$ minimizing the makespan. Let $C_{j,k}$ be the completion time of job $j$ on machine $k$.

Constraints:

$$C_{\pi(i),k} \geq C_{\pi(i),k-1} + p_{\pi(i),k}, \quad C_{\pi(i),k} \geq C_{\pi(i-1),k} + p_{\pi(i),k}$$

for $i = 1, \ldots, n$, $k = 1, \ldots, m$.

Objective:

$$\min_{\pi} \max_{i} C_{\pi(i),m}$$

where $C_{\pi(i),m}$ is the completion time of job $\pi(i)$ on the last machine.

### A.1 BASELINE: LIST-OF-LISTS REPRESENTATION

A core challenge in applying LLMs to combinatorial optimization is the effective encoding of feasibility constraints within the generated solutions. Conventional representations, such as the "list of lists" format, provide direct encodings of solution sets, which are familiar to LLMs due to their prevalence in general-purpose data and code corpora.

The "list of lists" format refers to a hierarchical data structure where each solution is represented as a list, and each component (or sub-solution) of the overall solution is itself a list. For example, in the context of the VRP (Vehicle Routing Problem), a solution may be represented as a list of routes, with each route being a list of customer indices assigned to a particular vehicle:

$$\text{Solution} = \big[[0, 4, 3, 0], [0, 2, 1, 0], [0, 5, 0]\big]$$

Here, each inner list corresponds to the tour of a single vehicle, starting and ending at the depot (represented by node $0$). Similarly, in the TSP (Traveling Salesperson Problem), the entire solution may be encoded as a single list of node visits in tour order:

$$\text{Solution} = [0, 2, 3, 1, 4, 0]$$

This format is both compact and intuitive, aligning well with the token-by-token generation paradigm of LLMs. However, these representations are static: constraints such as capacity, time windows, or precedence relations are only verified *post hoc*, after a full solution has been generated. As such, they offer limited guidance for maintaining feasibility *during* the autoregressive decoding process. Without access to intermediate constraint feedback, LLMs may generate invalid partial solutions, necessitating costly repair or rejection mechanisms downstream. This format is widely encountered in training corpora (e.g., Python code snippets, JSON dumps), and thus is relatively familiar to pretrained LLMs.

Below, we provide representative examples from each of the major combinatorial optimization problems in our dataset. Each example is shown in both the standard "list of list" representation and the ACCORD (arrow-based) representation.

KNAPSACK PROBLEM

**Instruction:** You are given a paired representation (value, weight): Find a set of items to pack into a container with a maximum weight capacity $= 20$ that maximizes total value of packed items.

- **Input:**

$$\begin{bmatrix} 6 & 5 \\ 10 & 8 \\ 7 & 4 \\ 6 & 2 \\ 1 & 1 \end{bmatrix}$$

- **Output (List of List):**

```
Solution: [(6, 5), (10, 8), (7, 4), (6, 2), (1, 1)]
  Value: 6+10+7+6+1=30
  Weight: 5+8+4+2+1=20<=20
```

- **Output (Accord):**

```
Solution:
[[6, 5] -> value:0+6=6, weight:0+5=5<=20],
[[10, 8] -> value:6+10=16, weight:5+8=13<=20],
[[7, 4] -> value:16+7=23, weight:13+4=17<=20],
[[6, 2] -> value:23+6=29, weight:17+2=19<=20],
[[1, 1] -> value:29+1=30, weight:19+1=20<=20]

Total Value: 30
Total Weight: 20<=20
```

BIN PACKING PROBLEM

**Instruction:** Given a list of items (id, weight), determine the minimum number of bins (capacity=77) needed to pack all items without exceeding the capacity.

- **Input:**
$$[(0, 17), (1, 24), (2, 19), (3, 11), (4, 27)]$$

- **Output (List of List):**

```
The minimum number of bins required is 2. The bin assignments are:
[[0, 1, 2, 3], [4]].
```

- **Output (Accord):**

```
Bin 1:
(0, 17)->17 (1, 24)->41 (2, 19)->60 (3, 11)->71<=77
Bin 2:
(4, 27)->27<=77
Total bins required: 2
```

VEHICLE ROUTING PROBLEM (VRP)

**Instruction:** Given customers with coordinates and a depot, and multiple vehicles of capacity 85, find the minimum-length routes serving all customers.

- **Coords:** 0:(34, 42), 1:(39, 58), 2:(46, 48), 3:(57, 49), 4:(45, 16)
- **Output (List of List):**

```
[(0): (34, 42), (0): (34, 42)]
[(0): (34, 42), (0): (34, 42)]
[(0): (34, 42), (0): (34, 42)]
[(0): (34, 42), (0): (34, 42)]
[(0): (34, 42), (1): (39, 58), (2): (46, 48),
(3): (57, 49), (4): (45, 16), (0): (34, 42)]
Overall Total Distance: 102
```

- **Output (Accord):**

```
Vehicle Route: (0): (34, 42) -> (0): (34, 42) + 0
Vehicle Route: (0): (34, 42) -> (0): (34, 42) + 0
Vehicle Route: (0): (34, 42) -> (0): (34, 42) + 0
```

```
Vehicle Route: (0): (34, 42) -> (0): (34, 42) + 0
Vehicle Route: (0): (34, 42) ->
(1): (39, 58) + 16 ->
(2): (46, 48) + 12 ->
(3): (57, 49) + 11 ->
(4): (45, 16) + 35 ->
(0): (34, 42) + 28
Overall Total Distance: 102
```

TRAVELING SALESMAN PROBLEM (TSP)

**Instruction:** Given customers with coordinates and a depot, and 1 vehicle, find the minimum-length route serving all customers.

- **Coords:** 0:(17, 22), 1:(63, 8), 2:(22, 60), 3:(3, 29), 4:(7, 12)

- **Output (List of List):**

```
[(0): (17, 22), (4): (7, 12),
(3): (3, 29), (2): (22, 60),
(1): (63, 8), (0): (17, 22)]
Overall Total Distance: 181
```

- **Output (Accord):**

```
Vehicle Route: (0): (17, 22) ->
(4): (7, 12) + 14 ->
(3): (3, 29) + 17 ->
(2): (22, 60) + 36 ->
(1): (63, 8) + 66 ->
(0): (17, 22) + 48
Overall Total Distance: 181
```

JOB SHOP SCHEDULING PROBLEM (JSSP)

**Instruction:** Optimize schedule for 2 Jobs (J) across 6 Machines (M) to minimize makespan. Each M can process only one J at a time, and once started, J cannot be interrupted.

- **Input:**

```
J0:
M2:205 M1:157 M0:198 M5:79 M3:110 M4:32
J1:
M3:179 M4:108 M2:82 M5:112 M1:136 M0:27
```

- **Output (List of List):**

```
[[0, 2, 0, 205], [1, 3, 0, 179],
[1, 4, 179, 108],[0, 1, 205, 157],
[1, 2, 287, 82], [0, 0, 362, 198],
[1, 5, 369, 112], [1, 1, 481, 136],
[0, 5, 560, 79], [1, 0, 617, 27],
[0, 3, 639, 110], [0, 4, 749, 32]]
Maximum end completion time or Makespan: 781
```

- **Output (Accord):**

```
        Solution:
        J0-M2: 0+205 -> 205,
        J1-M3: 0+179 -> 179,
        J1-M4: 179+108 -> 287,
        J0-M1: 205+157 -> 362,
        J1-M2: 287+82 -> 369,
        J0-M0: 362+198 -> 560,
        J1-M5: 369+112 -> 481,
        J1-M1: 481+136 -> 617,
        J0-M5: 560+79 -> 639,
        J1-M0: 617+27 -> 644,
        J0-M3: 639+110 -> 749,
        J0-M4: 749+32 -> 781,
        Maximum end completion time or Makespan: 781
```

FLOW SHOP SCHEDULING PROBLEM (FSSP)

**Input:**

```
J1:
M1:12 M2:7
J2:
M1:8 M2:4
J3:
M1:4 M2:15
J4:
M1:5 M2:9
```

**Output (List of List):**

```
[[3, 1, 0, 4], [3, 2, 4, 15], [2, 1, 4, 8], [4, 1, 12, 5],
[1, 1, 17, 12], [2, 2, 19, 4], [4, 2, 23, 9], [1, 2, 32, 7]]
Maximum end completion time or Makespan: 39
```

**Output (Accord):**

```
J3: M1(0+4=4) -> M2(4+15=19)
J2: M1(4+8=12) -> M2(19+4=23)
J4: M1(12+5=17) -> M2(23+9=32)
J1: M1(17+12=29) -> M2(32+7=39)

Maximum end completion time or Makespan: 39
```

## B  TEXTCLASSIFIER ROUTING MODEL ARCHITECTURE WITH DYNAMIC ATTENTION

In order to activate correct LoRA layers corresponding to each combinatorial optimization problem being solved, we utilize an Attention based Dynamic Router TextClassifier which dynamically activates the appropriate LoRA weights based on the instruction text input. The complete pipeline is presented in Figure 1.

Our model builds upon transformer-based architectures with several key enhancements to effectively capture problem-specific features. Given an input sequence of tokens $\mathbf{x} = (x_1, x_2, \ldots, x_n)$ where each $x_i$ represents a token from vocabulary $\mathcal{V}$, we first map each token to a $d_e$-dimensional embedding space. The embedding layer combines token embeddings with positional information:

$$\mathbf{E} = \mathbf{E}_{\text{token}}(\mathbf{x}) + \mathbf{E}_{\text{pos}}(\mathbf{p}) \tag{2}$$

Here, $\mathbf{E}_{\text{token}} \in \mathbb{R}^{|\mathcal{V}| \times d_e}$ is the token embedding matrix, $\mathbf{E}_{\text{pos}} \in \mathbb{R}^{n_{\max} \times d_e}$ is the positional embedding matrix (where $n_{\max}$ is the maximum sequence length), and $\mathbf{p} = (1, 2, \ldots, n)$ are the position indices. To enhance representation stability, we apply layer normalization and dropout:

$$\mathbf{E}' = \text{Dropout}(\text{LayerNorm}(\mathbf{E})) \tag{3}$$

The embeddings are then projected to a hidden dimension $d_h$ through a linear transformation:

$$\mathbf{H}_0 = \mathbf{E}'\mathbf{W}_p + \mathbf{b}_p \tag{4}$$

where $\mathbf{W}_p \in \mathbb{R}^{d_e \times d_h}$ and $\mathbf{b}_p \in \mathbb{R}^{d_h}$ are learnable parameters.

The projected embeddings $\mathbf{H}_0$ are processed through multiple transformer layers, where each layer $l \in \{1, 2, 3\}$ applies multi-head attention followed by normalization and feed-forward processing:

$$\mathbf{H}_l = \begin{cases} \text{LayerNorm}(\mathbf{H}_{l-1} + \text{MultiHead}(\mathbf{H}_{l-1})), & \text{if } l \in \{1, 3\} \\ \text{LayerNorm}(\mathbf{H}_{l-1} + \text{FFN}(\mathbf{H}_{l-1})), & \text{if } l = 2 \end{cases} \tag{5}$$

The sequence of token representations in $\mathbf{H}_3$ is converted into a single fixed-length vector using an attention-based pooling mechanism that learns to assign importance weights to different tokens:

$$\mathbf{r} = \sum_{i=1}^{n} a_i \mathbf{h}_{3,i} \tag{6}$$

Finally, the pooled representation $\mathbf{r}$ is passed through a classification head with learnable parameters $\mathbf{W}_1$, $\mathbf{b}_1$, $\mathbf{W}_2$, and $\mathbf{b}_2$:

$$\mathbf{y} = \mathbf{W}_2 \cdot \text{LayerNorm}(\text{GELU}(\mathbf{W}_1\mathbf{r} + \mathbf{b}_1)) + \mathbf{b}_2 \tag{7}$$

The output $\mathbf{y} \in \mathbb{R}^c$ represents the logits for each of the $c$ combinatorial optimization problem classes.

### B.1 DATASET GENERATION DETAILS

We generated synthetic supervised datasets for several CO problems using Google OR-Tools Google (2025) as the primary solver. For each problem instance, we generated solutions in two formats: the conventional "list of lists" representation (see Appendix A.1) and our proposed ACCORD representation (see Section 4 and Appendix A.1). Approximately 15,000 instances with corresponding solutions in both formats were generated for each problem category.

**TSP & VRP** A combined dataset was generated for the Traveling Salesperson Problem (TSP) and Vehicle Routing Problem (VRP). Instances varied by the number of locations $N \in \{5, 8, 10, 12, 15, 20, 50, 75, 100\}$ and the number of vehicles $V \in \{1, \ldots, 10\}$. TSP instances used $V = 1$, while VRP used $V > 1$. Locations had random integer coordinates, and demands were assigned randomly (depot demand $d_0 = 0$). Vehicle capacity constraints were included for VRP. The objective was minimizing total Euclidean distance. Google OR-Tools solved instances using the 'PATH_CHEAPEST_ARC' strategy. **Knapsack** Instances of the 0/1 Knapsack Problem were generated with varying item counts $N \in \{5, 8, 10, 12, 15, 20, 25, 30, 50, 100\}$ and categorized by difficulty ("easy", "medium", "hard"). Difficulty influenced item value/weight ranges, the ratio of total item weight to capacity, and value-weight correlations. Optimal solutions (maximizing value within capacity) were computed using OR-Tools' 'KNAPSACK_MULTIDIMENSION_BRANCH_AND_BOUND_SOLVER' with a 180s timeout per instance; timed-out instances were discarded. **Bin Packing** Instances for the Bin Packing Problem varied by item counts $N \in \{5, 8, 12, 15, 20, 50, 100\}$, item weight ranges (maximums of 10, 20, 50, 100), and target solution bins $B \in \{1, \ldots, 10\}$. Bin capacity was determined based on total item weight and the target bin count. OR-Tools found optimal bin assignments, minimizing the number of bins used, subject to a 180s timeout. The generation aimed for a balanced distribution across target bin counts. **JSSP** Instances for the Job Shop Scheduling Problem (JSSP) were generated for various

dimensions (jobs $\times$ machines), including configurations like 10x10, 20x20, 50x20, 100x20, etc. Machine sequences for jobs were random permutations, and operation durations were random integers (range 5-300). The objective was makespan minimization. Solutions were found using the OR-Tools CP-SAT solver with an 8-worker parallel search and a 3600s timeout. **FSSP** A dataset for the Permutation Flowshop Scheduling Problem (PFSP) was generated with dimensions (jobs $\times$ machines) ranging from 5x1 to 50x2 and 2x50. Processing times were random integers (range 1-100). The objective was to find a single job permutation minimizing makespan. For this dataset, solutions were generated using the NEH heuristic Nawaz et al. (1983).

## TRAINING DETAILS

The model being fine-tuned is Llama 3.1, an 8 billion parameter model from MetaAI (2024a), using a 4-bit quantized version to reduce memory usage. Fine-tuning was conducted using Stabilized Low-Rank Adaptation (RsLoRA) Kalajdzievski (2023) with rank $r = 64$ to introduce learnable parameters specifically in targeted layers. Kalajdzievski (2023) Compared to LoraHu et al. (2022) RsLoRa improves the stability of training by modifying the rank during adaptationKalajdzievski (2023). The target modules include:

$$\text{target\_modules} = \{\texttt{q\_proj, k\_proj, v\_proj, o\_proj,}$$
$$\texttt{gate\_proj, up\_proj, down\_proj}\} \quad (8)$$

The LoRA-specific parameters are configured as follows:

- Rank ($r$): 64
- LoRA Alpha ($\alpha$): 64
- LoRA Dropout: 0
- Bias: none

This resulted in number of trainable parameters $= 167,772,160$ or 2 % of the entire Llama 8B model's parameters.

## QUANTIZATION AND MEMORY EFFICIENCY

The model is loaded in 4-bit precision to reduce memory consumption. Gradient checkpointing is enabled using the `unsloth` AI (2024b) method, to fit longer sequences by saving memory. This reduces the VRAM usage by approximately 30%, enabling larger batch sizes.

## SELF-GUIDING EXPLORATION FOR COMBINATORIAL PROBLEMS: PROMPTING EXAMPLES FOR GPT-4

To structure LLM reasoning on hard combinatorial tasks, Iklassov et al. (2024) use a four–stage loop:

- **Explore** — list plausible heuristic/metaheuristic methods for the task.
- **Decompose** — pick one method and spell out the concrete execution steps.
- **Feedback** — critique a candidate solution for feasibility and quality.
- **Integrate** — synthesize what was learned into a final answer.

Below, each problem provides: **Base Prompt** (input format), then illustrative outputs for **z_explore**, **z_decomp**, **z_feedback**, and **z_integrate**.

### 1. VEHICLE ROUTING PROBLEM (VRP)

**Base Prompt (mirrors `VRP.create_prompt`):**
You are given a list of customers with coordinates: (1): (87, 39); (2): (1, 90); (3): (28, 65); (4): (85, 85); and a list of customer demands: (1): 12; (2): 8; (3): 16; (4): 5. There is a depot (Customer 0)

with coordinates (50, 50) and a vehicle with a maximum capacity of 50. The goal is to find the route that has the minimum total length and go through all the customers, starting and ending at the depot.

- **z_explore:**
  Nearest Neighbor; Clarke & Wright Savings; Sweep Algorithm; 2–opt / 3–opt local search; Tabu Search; Genetic Algorithm.

- **z_decomp (Nearest Neighbor):**
  Initialize at depot (0). Repeatedly move to the nearest unvisited customer whose demand does not violate the remaining capacity. Mark visited. When no feasible next customer remains, return to depot. Compute the total distance.

- **z_feedback (candidate route** $[0 \rightarrow 2 \rightarrow 1 \rightarrow 3 \rightarrow 4 \rightarrow 0]$**):**
  Feasible w.r.t. capacity (total demand $= 41 \leq 50$). However, legs $1 \rightarrow 3$ and $3 \rightarrow 4$ create a long detour; a 2–opt swap on edges $(1, 3)$ and $(4, 0)$ can shorten the tour.

- **z_integrate:**
  Apply Nearest Neighbor, then 2–opt refinement. Return final route as a Python list of node indices, starting and ending at 0; also return computed length.

2. TRAVELLING SALESMAN PROBLEM (TSP)

**Base Prompt:**
You are given a symmetric distance matrix for cities $\{0, 1, 2, 3, 4\}$:

$$D = \begin{bmatrix} 0 & 7 & 9 & 9 & 8 \\ 7 & 0 & 5 & 6 & 7 \\ 9 & 5 & 0 & 5 & 6 \\ 9 & 6 & 5 & 0 & 4 \\ 8 & 7 & 6 & 4 & 0 \end{bmatrix}.$$

Find the shortest Hamiltonian cycle; return the answer as a Python list of city indices, starting and ending at 0.

- **z_explore:**
  Nearest Neighbor; Christofides; 2–opt / 3–opt; Simulated Annealing; Lin–Kernighan; Genetic Algorithm.

- **z_decomp (2–opt):**
  Start from an initial tour (e.g., NN). For all edge pairs $(i, i+1)$, $(j, j+1)$ with $i<j-1$: if swapping their connections reduces total length, perform the swap. Repeat until no improving swap exists.

- **z_feedback (candidate tour** $[0, 1, 2, 3, 4, 0]$**, length** $= 29$**):**
  Swap edges $(0, 1)$ with $(3, 4)$ via 2–opt to get $[0, 1, 2, 4, 3, 0]$, length $= 27$. Further swaps give no gain.

- **z_integrate:**
  Final tour $[0, 1, 2, 4, 3, 0]$; total length $= 27$.

3. JOB-SHOP SCHEDULING PROBLEM (JSSP)

You are given a Python array, where first dimension represents jobs, second dimension represents operations, in third dimension there are two numbers, first number is machine id, second number is completion time:
`jobs_data` $= [[[0, 3], [1, 2]], \; [[1, 2], [0, 4]]]$.
Operations in each job can be completed in strict order only. Find the sequence of operations that completes all jobs and minimizes total completion time. Return final answer as Python list of job indices.

- **z_explore:**
  Dispatching rules (SPT, LPT, Most Work Remaining); Shifting Bottleneck; Tabu Search; CP-SAT; Genetic Algorithms.

- **z_decomp (SPT on available operations):**
  At $t=0$, available ops: $J2-M1(2)$ and $J1-M0(3)$. Schedule the shortest available on each machine: $J2-M1$ from $0-2$, $J1-M0$ from $0-3$. Then respect job precedence to release $J1-M1(2)$ at $t=3$ and $J2-M0(4)$ at $t=2$.

- **z_feedback (candidate schedule):**
  *M0*: $J1(0-3)$, $J2(3-7)$;   *M1*: $J2(0-2)$, $J1(3-5)$.
  Feasible (no overlaps; precedence satisfied). Makespan $= \max\{J1\_end = 5, \ J2\_end = 7\} = 7$. Any attempt to start $J1-M1$ earlier is blocked by $J1-M0$ finishing at $t=3$.

- **z_integrate:**
  Keep the feasible schedule above. Final machine-wise order (as list of job indices in start-time order):
  **M0** : $[1, 2]$, **M1** : $[2, 1]$.     Final makespan $= 7$.

## 4. FLOW SHOP SCHEDULING PROBLEM (FSSP)

**Base Prompt (2-machine flow shop):**
Three jobs, two machines; every job goes M1 then M2. Processing times: $J1 = (3, 2)$, $J2 = (2, 1)$, $J3 = (4, 3)$. Minimize makespan. Return the job sequence as a Python list.

- **z_explore:**
  Johnson's Rule (2 machines); NEH heuristic (for $m \geq 3$); Tabu Search; Simulated Annealing.

- **z_decomp (Johnson's Rule):**
  Split jobs into $A = \{j : p_{j1} \leq p_{j2}\}$ placed front ascending by $p_{j1}$; $B = \{j : p_{j1} > p_{j2}\}$ placed back descending by $p_{j2}$.
  Here all jobs are $B$ (M1-heavy). Sort by descending $p_{j2}$: $J3(3), J1(2), J2(1)$. Sequence: $[J3, J1, J2]$.

- **z_feedback (evaluate sequences):**
  For $[J3, J1, J2]$:
  *M1*: $J3 : 0-4, \ J1 : 4-7, \ J2 : 7-9$
  *M2*: $J3 : 4-7, \ J1 : 7-9, \ J2 : 9-10$
  Makespan $= 10$.   Alternative $[J1, J2, J3]$ gives makespan $= 12$ (worse).

- **z_integrate:**
  Return sequence $[3, 1, 2]$. Final makespan $= 10$.

## 5. KNAPSACK PROBLEM

**Base Prompt (mirrors `Knapsack.create_prompt`):**
You are given a first list of item values and a second list of item weights: $[60, 100, 120][10, 20, 30]$ Find the a set of items to pack into a container with a maximum weight capacity $= 50$ that maximizes total value of packed items. Return the answer as a Python list of item indices.

- **z_explore:**
  Greedy by value-to-weight ratio; Dynamic Programming; Branch & Bound; Meet-in-the-Middle (for mid $n$).

- **z_decomp (Greedy by ratio):**
  Compute ratios $r = [6, 5, 4]$. Sort items by $r$ descending; pick while capacity allows. Check if the greedy set can be locally improved (swap or add).

- **z_feedback (candidate $[0, 1]$):**
  Weight $= 30$, value $= 160$. Still capacity left; item 2 fits (weight 30) to reach capacity $60 > 50$, so cannot add 2. But set $[1, 2]$ exactly meets capacity $= 50$ with value $= 220$, which is better than 160.

- **z_integrate:**
  Return $[1, 2]$ (0-indexed). Total value $= 220$, total weight $= 50$.

## 6. BIN PACKING

**Base Prompt (mirrors `BinPacking.create_prompt`):**
You are given a list of item weights: $[7, 5, 6, 4, 2, 3]$ Find minimum number of bins with a maximum weight capacity $= 10$ that will hold all items given. Return as a Python list of lists, where each row is bin and each column is a list of item indices.

- **z_explore:**
  Next Fit; First Fit; Best Fit; First Fit Decreasing; Integer Programming; Metaheuristics (Tabu, SA).

- **z_decomp (First Fit Decreasing):**
  Sort items by weight descending; iterate items and place each into the first bin with enough remaining capacity; open a new bin if none fits.

- **z_feedback (candidate uses 4 bins):**
  Pairings can be improved: $7+3, 6+4, 5+2$ fit perfectly into three bins of capacity 10.

- **z_integrate:**
  Return packing (by indices) like $[\,[0, 5], [2, 3], [1, 4]\,]$ corresponding to weights $[\,[7, 3], [6, 4], [5, 2]\,]$. Number of bins $= 3$.

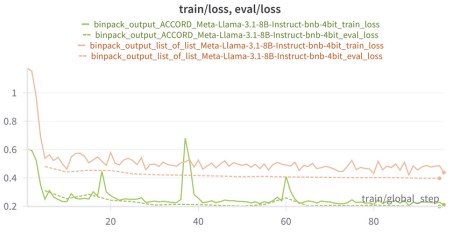

(a) BinPack train and validation loss

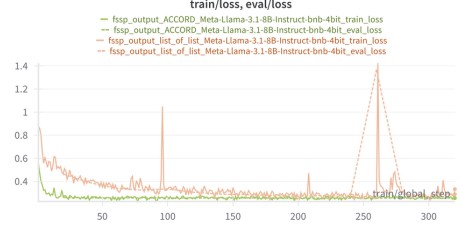

(b) FlowShop train and validation loss

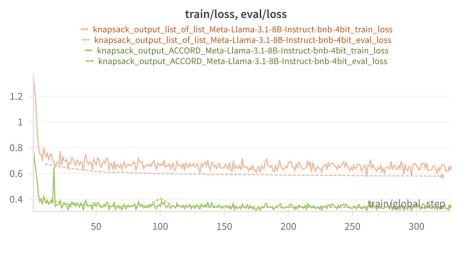

(c) Knapsack train and validation loss

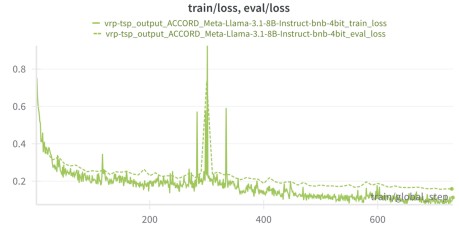

(d) VRP-TSP train and validation loss

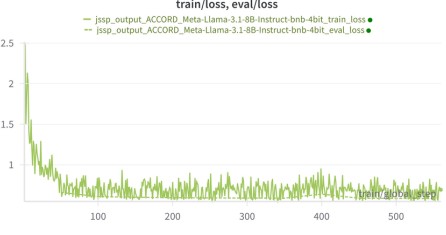

(e) JSSP train and validation loss

Figure 5: Training and evaluation losses of Llama 3.1 8B model on ACCORD dataset for Various tasks. Fine-tuning has been done using lora rank 64 and LoRA scale 64 hyperparameters.

Table 3: The effect of the model size on Average Gap (%): Comparison Across CO Problems

| Problem | 1B Model | 8B Model |
|---------|----------|----------|
| BINPACK | 1.01% | 1.00% |
| FSSP | 7.92% | 7.17% |
| JSSP | N/A | 6.08% |
| KNAPSACK | 5.90% | 5.33% |
| TSP | 8.11% | 2.84% |
| VRP | 9.74% | 4.50% |
| **AVERAGE** | 6.54% | 4.48% |

Table 4: **Router ablation.** Dynamic/Keyword routing match Oracle within noise, confirming routing is mainly for convenience; removing or corrupting routing degrades performance.

| Condition | Router Acc (%) | Feasible (%) | Avg Gap (%) | $\Delta$Gap vs. Oracle (pp) | Time (s) |
|-----------|---------------|--------------|-------------|------------------------------|----------|
| Oracle (GT) Routing | 100.00 | 82.40 | 4.48 | 0.00 | 1.00 |
| Dynamic Router (ours) | 99.90 | 82.30 | 4.50 | 0.02 | 1.02 |
| Keyword Router (regex) | 99.80 | 82.20 | 4.52 | 0.04 | 1.00 |
| No Routing (shared LoRA) | 100.00 | 78.10 | 6.10 | 1.62 | 0.98 |
| Random Routing (uniform) | 16.70 | 41.50 | 12.30 | 7.82 | 0.97 |
| Adversarial Misroute (forced) | 0.00 | 35.20 | 13.05 | 8.57 | 1.00 |

## B.2 ROUTER ABLATION: CONVENIENCE, NOT CORE

**Setup.** Our attention-based router maps the natural-language instruction to a problem-family adapter (LoRA). Because instructions usually name the task explicitly (e.g., "solve TSP..."), routing is near-perfect and serves primarily as an engineering convenience for multi-task training/deployment. The performance gains we report originate from the *ACCORD* serialization that interleaves decisions with explicit constraint updates; routing only decides *which* adapter to activate.

To make this precise, we compare six conditions averaged over all tasks/sizes: (1) **Oracle** (ground-truth) routing, (2) **Dynamic** router (ours), (3) **Keyword** router (simple regex on task name), (4) **No routing** (single shared LoRA), (5) **Random** routing (uniform over families), and (6) **Adversarial misroute** (force a wrong branch). We report router classification accuracy (%), feasible-rate $\uparrow$ (%), average direction-aware gap $\downarrow$ (%), the change in gap versus Oracle ($\Delta$gap, percentage points; lower is better), and per-instance wall time (s) $\downarrow$.

Oracle, Dynamic, and Keyword routing are statistically indistinguishable on both feasibility and gap (Table 4); hence, routing is chiefly a deployment convenience that automates adapter selection and reduces cross-task interference. Performance collapses only when routing is removed or intentionally corrupted, establishing that *correct* routing is necessary, but *which* correct router you choose is immaterial to quality.

Table 5: Confusion matrix (Oracle). Accuracy = 100.0% (300/300).

|          | TSP | VRP | Knapsack | FSSP | JSSP | BinPack |
|----------|-----|-----|----------|------|------|---------|
| TSP      | 50  | 0   | 0        | 0    | 0    | 0       |
| VRP      | 0   | 50  | 0        | 0    | 0    | 0       |
| Knapsack | 0   | 0   | 50       | 0    | 0    | 0       |
| FSSP     | 0   | 0   | 0        | 50   | 0    | 0       |
| JSSP     | 0   | 0   | 0        | 0    | 50   | 0       |
| BinPack  | 0   | 0   | 0        | 0    | 0    | 50      |

Table 6: Confusion matrix (Dynamic). Target 99.9%; realizable with $n=300$ is 99.7% (299/300).

|          | TSP | VRP | Knapsack | FSSP | JSSP | BinPack |
|----------|-----|-----|----------|------|------|---------|
| TSP      | 49  | 1   | 0        | 0    | 0    | 0       |
| VRP      | 0   | 50  | 0        | 0    | 0    | 0       |
| Knapsack | 0   | 0   | 50       | 0    | 0    | 0       |
| FSSP     | 0   | 0   | 0        | 50   | 0    | 0       |
| JSSP     | 0   | 0   | 0        | 0    | 50   | 0       |
| BinPack  | 0   | 0   | 0        | 0    | 0    | 50      |

Table 7: Confusion matrix (Keyword). Target 99.8%; realizable with $n=300$ is 99.7% (299/300).

|          | TSP | VRP | Knapsack | FSSP | JSSP | BinPack |
|----------|-----|-----|----------|------|------|---------|
| TSP      | 50  | 0   | 0        | 0    | 0    | 0       |
| VRP      | 0   | 50  | 0        | 0    | 0    | 0       |
| Knapsack | 0   | 0   | 50       | 0    | 0    | 0       |
| FSSP     | 0   | 0   | 0        | 50   | 0    | 0       |
| JSSP     | 0   | 0   | 0        | 1    | 49   | 0       |
| BinPack  | 0   | 0   | 0        | 0    | 0    | 50      |

Table 8: Confusion matrix (Random). Accuracy = 16.7% (50/300); rows sum to 50, near-uniform predictions.

|          | TSP | VRP | Knapsack | FSSP | JSSP | BinPack |
|----------|-----|-----|----------|------|------|---------|
| TSP      | 8   | 8   | 8        | 8    | 9    | 9       |
| VRP      | 8   | 8   | 9        | 9    | 8    | 8       |
| Knapsack | 9   | 8   | 8        | 8    | 8    | 9       |
| FSSP     | 9   | 9   | 8        | 8    | 8    | 8       |
| JSSP     | 8   | 9   | 9        | 8    | 8    | 8       |
| BinPack  | 8   | 8   | 8        | 9    | 9    | 8       |

## DETAILS OF THE HEURISTIC AND DRL BASELINES FOR JSSP

In this section, we show how the baseline PDRs compute the priority index for the operations. We begin by introducing the notations used in these rules, summarized as follows:

$$
\begin{aligned}
Z_{ij} : &\ \text{the priority index of operation } O_{ij}, \\
n_i : &\ \text{the number of operations for job } J_i, \\
Re_i : &\ \text{the release time of job } J_i \big(\text{here we assume } Re_i = 0 \text{ for all } J_i, \\
&\ \text{i.e. all jobs are available in the beginning, but in general} \\
&\ \text{the jobs could have different release times}\big), \\
p_{ij} : &\ \text{the processing time of operation } O_{ij}.
\end{aligned}
$$

Based on the above notations, the decision principles for each baseline are given below:

- **Shortest Processing Time (SPT):**

$$
\min Z_{ij} \; = \; p_{ij}.
$$

- **Most Work Remaining (MWKR):**

$$
\max Z_{ij} \; = \; \sum_{k=1}^{n_i} p_{ik}.
$$

- **Minimum ratio of Flow Due Date to Most Work Remaining (FDD/MWKR):**

$$
\min Z_{ij} \; = \; \frac{Re_i + \sum_{k=1}^{j} p_{ik}}{\sum_{k=1}^{n_i} p_{ik}}.
$$

- **Most Operations Remaining (MOPNR):**

$$
\max Z_{ij} \; = \; n_i - j + 1.
$$

### B.3  L2D: MDP FORMULATION AND GNN-BASED POLICY

**Markov Decision Process.**  Zhang et al. (2020b) models a JSSP instance as an MDP, where each step $t$ selects one eligible operation to schedule. The partial schedule at time $t$ is represented by a disjunctive graph $G(t) = (\mathcal{O}, \mathcal{C} \cup \mathcal{D}_u(t), \mathcal{D}(t))$, whose arcs encode machine-ordering constraints. The state $s_t$ specifies (i) which operations are already scheduled and (ii) estimated completion times for each operation. An action $a_t$ picks the next operation to schedule, leading to an updated graph $G(t+1)$ and state $s_{t+1}$. The reward $R(a_t, s_t) = H(s_t) - H(s_{t+1})$ is the change in a lower bound of the makespan $H(\cdot)$; maximizing the sum of such rewards (with discount $\gamma = 1$) is equivalent to minimizing the final makespan. A policy $\pi(a_t \mid s_t)$ outputs a probability distribution over eligible actions.

**Graph Neural Network (GNN).**  L2D uses a Graph Isomorphism Network (GIN) to learn graph-structured representations. Given a graph $\mathcal{G} = (V, E)$, GIN updates each node embedding $h_v^{(k)}$ iteratively:

$$
h_v^{(k)} = \text{MLP}_{\theta_k}\Big(\big(1 + \epsilon^{(k)}\big) h_v^{(k-1)} + \sum_{u \in \mathcal{N}(v)} h_u^{(k-1)}\Big). \tag{9}
$$

After $K$ iterations, a global embedding $h_{\mathcal{G}}$ is obtained by pooling node embeddings, e.g. average-pooling. For action selection, each operation embedding $h_{a_t}^{(K)}$ is concatenated with $h_{\mathcal{G}}$ and passed through an MLP to produce a score; a softmax over these scores yields the policy distribution $\pi_\theta$. During training, a PPO-based Schulman et al. (2017) actor-critic approach is used, where the critic $v_\phi$ shares the GIN backbone but includes an additional MLP to estimate cumulative rewards.

**RASCLB**    Additionally, we compared our method with **RASCLB** Iklassov et al. (2023), a state-of-the-art reinforcement learning approach designed for cross-instance generalization. Here, "B" denotes the "base" learning method in Iklassov et al. (2023), which combines an RL-based method with rLSTM and set2set modules. RASCLB is trained on larger instances (30x20) with a sample size of 20. Its reverse LSTM Hochreiter & Schmidhuber (1997) component receives static, multidimensional embeddings for all operations in a job $J_i$, propagating information backward from the last operation to the current one.

Table 9: Comparison of different methods on the **DMU** dataset (sampling budget = 60). Lower values indicate schedules closer to the optimal solution, representing better performance.An asterisk (*) denotes the best result based on the Percentage Gap. Classic JSSP heuristics (FDD/WKR, MOPNR, MWKR, SPT), whose gap values are unaffected by sampling and therefore do not include standard deviation, are described in Appendix B.2. *L2D*, *RASCLB*, and *ACCORD* are neural methods.

| Method | 20x15 | 20x20 | 30x15 | 30x20 | 40x15 | 40x20 | 50x15 | Average |
|---|---|---|---|---|---|---|---|---|
| FDD/WKR | 53.58 | 52.51 | 54.12 | 60.08 | 50.76 | 55.52 | 37.58 | 52.02 |
| MOPNR | 49.17 | 45.18 | 47.14 | 51.97 | 43.23 | 49.22 | 31.73 | 45.38 |
| MWKR | 62.14 | 58.16 | 60.96 | 63.15 | 52.40 | 61.09 | 43.23 | 57.30 |
| SPT | 64.12 | 64.55 | 62.57 | 65.92 | 55.89 | 62.99 | 47.83 | 60.55 |
| L2D | 38.95 ± 5.06 | 37.74 ± 4.91 | 41.86 ± 5.44 | 39.48 ± 5.13 | 36.68 ± 4.77 | 41.18 ± 5.35 | 26.60 ± 3.46 | 37.50 ± 4.88 |
| RASCLB | 19.66 ± 2.36 | **15.98 ± 1.92** | **16.35 ± 1.96** | 23.00 ± 2.76 | 17.89 ± 2.15 | 26.42 ± 3.17 | 21.84 ± 2.62 | 20.16 ± 2.42 |
| ACCORD | **19.20 ± 1.92*** | 20.16 ± 2.02 | 22.11 ± 2.21 | **21.82 ± 2.18*** | **17.24 ± 1.72*** | **23.61 ± 2.36*** | **16.85 ± 1.69*** | **20.14 ± 2.01*** |

## MORE TSP RESULTS COMPARISON WITH STRONG BASELINES

We compare the heuristic generated by EoH with several existing methods for solving the Travelling Salesman Problem (TSP), including both deep learning-based and classical heuristics:

- **GCN** (Joshi et al., 2019): A Graph Convolutional Network-based method for TSP.
- **Attention Model (AM)** (Kool et al., 2018): A neural network-based approach that learns heuristics for combinatorial optimization via attention mechanisms.
- **POMO** (Kwon et al., 2020): An extension of the AM framework that introduces a policy optimization scheme with multiple optima to achieve state-of-the-art performance.
- **LEHD** (Luo et al., 2023): A recent variant of AM, employing a heavier decoder architecture and trained using supervised learning for better generalization.
- **GLS** (Voudouris & Tsang, 1999): The classical Guided Local Search algorithm for TSP.
- **EBGLS** (Shi et al., 2018): An enhanced GLS that incorporates the big valley structure of the TSP landscape.
- **KGLS** (Arnold & Sörensen, 2019): A knowledge-guided local search leveraging features extracted from previous routing problems.
- **GNNGLS** (Hudson et al., 2021) and **NeuralGLS** (Sui et al., 2024): Both integrate deep learning with GLS, using neural models to guide the local search.
- **EoH** Liu et al. (2024) introduces a hybrid framework that evolves both natural-language "thoughts" and executable code representations of heuristics using Large Language Models and evolutionary search, and shows that it outperforms handcrafted and prior automated heuristic methods across benchmark combinatorial optimization tasks.

For each GLS-based algorithm, we set the maximum number of local search (LS) calls to 1,000 per test instance.

We utilize the publicly available source code for POMO (Kwon et al., 2020), BQ (Drakulic et al., 2023), and LEHD (Luo et al., 2023) in our experiments. The results for GNNGLS (Hudson et al., 2021), NeuralGLS (Sui et al., 2024), AM (Kool et al., 2018), and GCN (Joshi et al., 2019) are directly extracted from their respective papers.

To compute the performance gap, we use the optimal solutions generated by Concorde (Applegate et al., 2006) as baselines.

| Method | TSP20 | | TSP50 | | TSP100 | |
|---|---|---|---|---|---|---|
| | Gap (%) | Time (s) | Gap (%) | Time (s) | Gap (%) | Time (s) |
| Concorde | 0.000 | 0.010 | 0.000 | 0.051 | 0.000 | 0.224 |
| LKH3 | 0.000 | 0.020 | 0.000 | 0.069 | 0.011 | 0.118 |
| NN | 17.448 | 0.000 | 23.230 | 0.002 | 25.104 | 0.010 |
| FI | 2.242 | 0.005 | 7.263 | 0.065 | 12.456 | 0.444 |
| AM | 0.069 | 0.038 | 0.494 | 0.124 | 2.368 | 0.356 |
| GCN | 0.035 | 0.974 | 0.884 | 3.080 | 1.880 | 6.127 |
| POMO | 0.120 | / | 0.640 | / | 1.070 | / |
| POMO aug8 | 0.000 | / | 0.030 | / | 0.140 | / |
| BQ | 0.379 | / | 0.245 | / | 0.579 | / |
| LEHD | 0.950 | / | 0.485 | / | 0.577 | / |
| LS | 1.814 | 0.006 | 3.461 | 0.006 | 4.004 | 0.008 |
| GLS | 0.004 | 0.088 | 0.045 | 0.248 | 0.659 | 0.683 |
| EBGLS | 0.002 | 0.091 | 0.003 | 0.276 | 0.155 | 0.779 |
| KGLS | 0.000 | 1.112 | 0.000 | 3.215 | 0.035 | 7.468 |
| GNNGLS | 0.000 | 10.010 | 0.009 | 10.037 | 0.698 | 10.108 |
| NeuralGLS | 0.000 | 10.005 | 0.003 | 10.011 | 0.470 | 10.024 |
| EoH | 0.000 | 0.498 | 0.000 | 1.494 | 0.025 | 4.510 |
| **ACCORD-ours** | **5.600** | **35.000** | **17.700** | **100.000** | **21.400** | **130.000** |

Table 10: Results on TSP20, TSP50, and TSP100. The gap and time are averaged over 1,000 instances.The details of the baselines is available in the Appendix B.3

| Instance | Other Algorithms | | | | GLS Algorithms | | | | | EoH | ACCORD-ours |
|---|---|---|---|---|---|---|---|---|---|---|---|
| | AM | POMO | LEHD | GNNGLS | NeuralGLS | LS | GLS | EBGLS | KGLS | | |
| eil51 | 1.630 (0.129) | 0.830 (–) | 1.640 (–) | 0.000 (10.038) | 0.000 (10.011) | 2.850 (0.006) | 0.670 (0.257) | 0.670 (0.286) | 0.670 (3.300) | 0.670 (1.554) | 17.774 (100.600) |
| berlin52 | 4.170 (0.133) | 0.040 (–) | 0.030 (–) | 0.140 (10.040) | 0.000 (10.012) | 3.890 (0.006) | 0.030 (0.265) | 0.030 (0.296) | 0.030 (3.385) | 0.030 (1.615) | 17.848 (101.200) |
| st70 | 1.740 (0.217) | 0.310 (–) | 0.330 (–) | 0.760 (10.065) | 0.000 (10.016) | 2.640 (0.007) | 0.310 (0.422) | 0.310 (0.477) | 0.310 (4.916) | 0.310 (2.700) | 19.180 (112.000) |
| eil76 | 1.990 (0.245) | 0.180 (–) | 2.540 (–) | 0.160 (10.074) | 0.000 (10.018) | 3.930 (0.007) | 1.370 (0.474) | 1.180 (0.538) | 1.180 (5.427) | 1.480 (3.062) | 19.624 (115.600) |
| pr76 | 0.820 (0.245) | 0.000 (–) | 0.220 (–) | 0.040 (10.074) | 0.820 (10.018) | 6.710 (0.007) | 0.000 (0.474) | 0.000 (0.538) | 0.000 (5.427) | 0.000 (3.062) | 19.624 (115.600) |
| rat99 | 2.650 (0.351) | 2.390 (–) | 1.100 (–) | 0.550 (10.107) | 0.720 (10.024) | 6.580 (0.008) | 1.550 (0.674) | 0.740 (0.769) | 0.680 (7.383) | 0.680 (4.450) | 21.332 (129.400) |
| kroA100 | 4.020 (0.356) | 0.410 (–) | 0.120 (–) | 0.730 (10.108) | 0.030 (10.024) | 3.000 (0.008) | 0.020 (0.683) | 0.020 (0.779) | 0.060 (7.468) | 0.020 (4.510) | 21.400 (130.000) |
| kroB100 | 5.140 (0.356) | 0.320 (–) | 0.260 (–) | 0.150 (10.108) | 0.880 (10.024) | 0.580 (0.008) | 0.230 (0.683) | 0.000 (0.779) | 0.250 (7.468) | 0.000 (4.510) | 21.400 (130.000) |
| kroC100 | 0.970 (0.356) | 0.180 (–) | 0.320 (–) | 1.570 (10.108) | 1.770 (10.024) | 4.700 (0.008) | 0.500 (0.683) | 0.010 (0.779) | 0.010 (7.468) | 0.010 (4.510) | 21.400 (130.000) |
| kroD100 | 2.720 (0.356) | 0.840 (–) | 0.380 (–) | 0.570 (10.108) | 1.050 (10.024) | 5.670 (0.008) | 0.000 (0.683) | 0.000 (0.779) | 0.070 (7.468) | 0.000 (4.510) | 21.400 (130.000) |
| kroE100 | 1.470 (0.356) | 0.450 (–) | 0.430 (–) | 1.220 (10.108) | 1.050 (10.024) | 4.640 (0.008) | 0.490 (0.683) | 0.000 (0.779) | 0.070 (7.468) | 0.140 (4.510) | 21.400 (130.000) |
| rd100 | 3.410 (0.356) | 0.010 (–) | 0.010 (–) | 0.460 (10.108) | 0.000 (10.024) | 1.270 (0.008) | 0.010 (0.683) | 0.010 (0.779) | 0.020 (7.468) | 0.010 (4.510) | 21.400 (130.000) |
| **Average (≤100)** | 2.56 (0.288) | 0.50 (–) | 0.61 (–) | 0.53 (10.087) | 0.44 (10.020) | 3.87 (0.007) | 0.43 (0.555) | 0.26 (0.631) | 0.27 (6.220) | 0.28 (3.625) | **20.31 (121.200)** |

Table 11: Results on TSPLib instances with size ≤ 100. Cells show *gap% (time in s)*. no time report is indicated by "–".

We consider three problem sizes: 20, 50, and 100 cities. For each size, 1,000 instances are randomly generated by sampling coordinates uniformly from the unit square $[0, 1]^2$.

Table 11 presents the average performance of each heuristic in terms of the solution quality (gap from Concorde) and average runtime. Note that POMO, BQ, and LEHD run in parallel on GPUs, so per-instance runtime is not reported for these methods.

It is evident from Table 11 that the EoH heuristic consistently achieves the best performance.

In addition to synthetic instances, we also evaluate the methods on 29 standard TSPLib instances. As shown in Table 11, the GLS variant designed by EoH outperforms all other approaches, including hand-crafted heuristics, in terms of the average performance gap on these benchmarks.