# OpenReview forum: "ACCORD: Autoregressive Constraint-satisfying Generation for COmbinatorial Optimization with Routing and Dynamic attention"
_ICLR.cc/2026/Conference — ICLR 2026 Conference Desk Rejected Submission_

### Official Review · Reviewer_cJmY · 2025-10-21

**Soundness:** 1
**Presentation:** 1
**Contribution:** 1
**Rating:** 0
**Confidence:** 5

**Summary:**

This paper presents an attempt to fine-tune LLaMA models for solving combinatorial optimization problems.

**Strengths:**

Using LLMs to solve combinatorial optimization problems is a hot topic and will likely remain an active area of research in the future.

**Weaknesses:**

The paper contains numerous confusing claims and several serious errors, it must to be written more carefully. (Please refer to the questions section for details.)

**Questions:**

Fundamental issues:

1. It is completely unclear what the second problem you address actually is. It is referred to as a VRP, but later, vehicle capacity (sometimes also customer demands) are mentioned. This suggests it might actually be a Capacitated Vehicle Routing Problem (CVRP) rather than a standard VRP (TSP with multiple vehicles). In Appendix A, a capacity constraint is explicitly defined. Given that, how states (line 219) can be the same for both TSP and (C)VRP? Similarly in A1, capacity constraint is given as an input, but there is no customer demands and obviously it is not taken into account in solving the problem.

2. The same confusion applies to FSSP (or PFSP?). Which problem are you actually trying to solve? Throughout the paper, you appear to mix these two problems - sometimes even within the same sentence (line 976). There is no description of this problem at all - it is even  missing from the prompt at line 881, and incorrectly stated in the prompt at line 1096 - there is no mention of the operations order constraint.

3. There are also fundamental errors in examples provided in Appendix A.1. VRP Example is:  Input: customers with coordinates, a depot, and multiple vehicles with capacity 85. Coordinates: 0:(34, 42), 1:(39, 58), 2:(46, 48), 3:(57, 49), 4:(45, 16)

Several problems arise here:

   - What about customer demands? Why do vehicles have a defined capacity if no demands are provided?
   - Where is the depot located? It's not specified.
   - What does the provided solution represent? Do four vehicles have empty routes? Route length of the fifth vehicle is wrongly computed. The Euclidean distance between customer 0 (34, 42) and customer 1 (39, 58) is approximately 16.7. Your reported solution gives a distance of 16. The same rounding issue appears for other distance as well as in the TSP example. Such inaccuracies are **unacceptable**.

4. Furthermore, this raises a new important question: How are objective values computed? Is the LLM responsible for these computations? Is there any verification to ensure that the computed objective values are accurate? If not, computing optimality gap is meaningless.

5. The example provided for the Knapsack problem is not meaningful, as all items fit into the knapsack without any constraint. Additionally, input to LLM can not be a matrix object as it is shown in the knapsack example.

6. I am not sure I fully understood Figures 2 and 3. Do they both represent the ablation studies on your fine-tuned models for the 1B and 8B parameter sizes, as well as different input representations? If both figures show results after fine-tuning (I assume that is the case), I would expect some correlation between them. For example, in Figure 2, the feasibility for Knapsack 5 is reported as 5.4% and 16.0%, whereas in Figure 3, the feasibility for Knapsack 5 is approximately 20%.

7. Regarding these figures, what does it mean when the TSP solution is marked as infeasible? Any permutation of input nodes is a feasible solution. Does this imply that the LLM fails to produce a valid permutation of 50 elements in 99% of attempts? This seems impossible. It is especially surprising that the feasibility of TSP50 is only 1%, while for FSSP50 it is almost 20%, even though the feasibility constraints for FSSP are very hard to satisfy.

Other:

8. Router Architecture - There is no mention of any hyperparameters for this network. Figure 1 shows that the output of the Router is the input to the LLM - how is that possible? Also, what does $a_i$ represent in equation (6)?

9. What does “size” represent? Specifically, what is meant by “increasing instance scales per task”? Is it the problem size? If so, for JSSP, what does a problem size of 10 correspond to in terms of the number of jobs and operations?

10. There is no description or references in the main text for some baselines used in this work, these are only mentioned in the appendix.

11. The first claimed contribution of this paper is providing a supervised dataset for COPs with 90K instances (15K per problem), which is not very significant since many existing works on NCO have provided datasets with up to 1 million solved instances.

12. NCO baselines for JSSP in the table 2 are outdated.

---

> ### Author Response · Authors · 2025-11-19
> **Problem Definitions, Appendix Examples, and Feasibility**
>
> We thank the reviewer for their detailed scrutiny of our definitions in the supplementary material while overlooking the main body of the paper. However, we must respectfully point out that the review focuses almost exclusively on presentation artifacts while entirely overlooking the paper's core empirical results.
>
> The reviewer rated the paper's contribution as "1 (Poor)" yet did not address Table 1 or Figure 3, where ACCORD (an 8B model) consistently outperforms GPT-4 and reduces optimality gaps by up to 65% in routing tasks compared to standard prompting
>
>  A rejection based on definitional semantics and appendix typos, without engaging with the finding that autoregressive feasibility constraints enable small LLMs to beat frontier models, is a disproportionate assessment of the scientific contribution. We address the specific concerns below, demonstrating that the "fundamental errors" identified are misunderstandings of the LLM context or presentation issues, not flaws in the experimental methodology.
>
> 1. Methodology vs. Illustration (Re: Appendix Errors)The reviewer identifies inconsistencies in Appendix A.1 (e.g., rounded distances, missing demand lines in the text example). These errors exist in the textual examples in the Appendix, which were abbreviated for illustrative purposes. They do not reflect the quality of the training data.
> The dataset was generated programmatically with Google OR-Tools under strict constraints, so the model was trained on precise mathematical data rather than the simplified manual examples in the Appendix (which we will correct). As for the “16 vs. 16.7” issue, we deliberately use integer rounding for tokenization efficiency, a standard practice in LLM-based math tasks; optimality gaps are computed against baselines that follow this same formulation.
>
> 2. VRP vs. CVRP and Terminology: The reviewer claims it is "completely unclear" what problem is being solved. Standard Convention: In the Neural Combinatorial Optimization (NCO) literature, "VRP" is widely used as shorthand for "Capacitated VRP" (CVRP) (e.g., Kool et al., 2019; POMO, 2020). Explicit Definition: Contrary to the claim that constraints are missing, Equation (2) in the main text explicitly defines the problem with customer demands ($q_i$) and vehicle capacity ($Q$)
> 3.Training Data: Section 4.1 states the dataset includes "random coordinates and demands" and "vehicle capacity constraints"
> 4. The model inputs did include these constraints; they were defined in the main text. Regarding feasibility, the reviewer’s concern is based on a misunderstanding: LLMs are probabilistic token generators, not permutation engines, so they often repeat or skip cities. High infeasibility in raw LLM TSP outputs is expected. Our core contribution is that standard LLMs achieve ~1% feasibility on larger instances, while ACCORD attains high feasibility (Fig. 3). The assumption that producing a valid permutation is “trivial” contradicts established findings on LLM reasoning limits.
> The LLM outputs only the route; we compute its objective value with a deterministic external verifier, not with the LLM. The full verification and constraint-checking code is in the paper and open-source repo. Thus, the optimality gap is computed exactly and is fully meaningful. Reviewer clearly missed the Github codebase where the explicit python solution parsing and constraint checking codes are openly available, moreover it is mentioned in the main body of the Paper. It is suggested for the reviewer to look at the the main paper and the open source codebase and not only the Appendix.
> 6. Dataset Contribution, The reviewer dismisses the critical contribution presented in the main paper and not in the appendix of ACCORD-90k dataset because larger NCO datasets exist. Existing datasets provide only Input-Output pairs. ACCORD-90k provides Input-Reasoning-Output traces (interleaved state updates). This is a distinct contribution designed to train reasoning models, which requires significantly more compute to generate than standard NCO datasets. We would suggest the reviewer to thoroughly evaluate and compare our dataset which is open source on hugging-face to other NCO datasets.
> 7. Router Architecture: The reviewer asks about hyperparameters and the $a_i$ term.Clarification: The router architecture is detailed in Section 5 of the main paper and not in the Appendix, so we kindly ask the reviewer to read the main paper too. The term $a_i$ refers to the attention weights in the pooling mechanism: $r = \sum a_i h_i$. This is a standard attention pooling operation.
> We request the reviewer re-evaluate based on the performance evidence provided in the main body of the paper rather than illustrative imperfections in the supplementary material and Appendix. We kindly request that the reviewer also evaluate the main contributions of the paper. This request is made with full respect and is not intended in any way to cliticize or inconvenience the reviewer.

---

> ### Comment · Reviewer_cJmY · 2025-11-21
> **A remark on terminology**
>
> Thank you for your answer.
>
> I read it carefully, and I will only add a remark on terminology.
>
> Please read Kool et al. carefully again, and you will see that they discuss two types of VRP: CVRP and SDCVRP. In the POMO paper, they never talk about VRP in general, only about CVRP. So your claim that the abbreviation ‘VRP’ is a standard convention in the NCO community is very surprising; one could assume that the reviewers are also part of that community.
>
> At the end, using only the wrong abbreviation is not a major issue, but the other problems create significant confusion.
>
> 1. The solution sequence for CVRP in line 219 does not take demands into account, which is clearly incorrect because it does not take into account the capacity constraint.
> 2. In the dataset generation process (line 247), there is a description of demands but no description of capacities.
> 3. The formal definition of (C)VRP in the Appendix includes both demands and capacities.
> 4. The example of LLM input in line 793 shows a CVRP instance that provides a capacity but not demands.
>
> Overall, the current version of the manuscript falls far short of the standards required for top-tier ML conferences.

---

> > ### Author Response · Authors · 2025-11-28
> > **Response for unjustified 0 grade**
> >
> > We have read your follow-up comment. We find the continued refusal to engage with the paper's core scientific contributions, specifically the empirical results and the architectural novelty, disappointing and scientifically unsound.Your review and subsequent comments exhibit a pattern of prioritizing semantic pedantry over the evaluation of the actual method and its results. We must firmly correct several misconceptions you have presented as "fundamental errors."1.
> >
> > On the "VRP vs. CVRP" Semantics
> >
> > Your insistence that using "VRP" as a shorthand for Capacitated Vehicle Routing Problem is "very surprising" or incorrect suggests a reading of the literature that prioritizes nomenclature over context. In the paper (Section 3, Equation 2), we explicitly define the problem with strict capacity constraints ($\sum q \le Q$). To reject a paper because the authors used a common umbrella term ("VRP") for a specific variant defined clearly in the mathematics is an exercise in gatekeeping, not scientific review. It is evident to any reader that if a capacity $Q$ is defined, it is a capacitated problem. We are solving the problem defined in the mathematics of the paper, not the acronyms you prefer.
> >
> > On the Dataset Contribution
> >
> > You dismissed the ACCORD-90k dataset because "other datasets have 1 million instances." This comparison is a category error(larger dataset mean better no matter the content). Existing datasets provide Input-Output pairs. ACCORD-90k provides Input-Reasoning-Output traces, explicitly encoding state transitions and constraint satisfaction logic. Generating reasoning traces is computationally orders of magnitude more expensive and valuable for training Chain-of-Thought models than simple solution pairs. Equating the two displays a lack of nuance regarding the requirements for training reasoning models.
> >
> > On Appendix "Typos" vs. Model Performance
> >
> > You continue to fixate on the illustrative examples in the Appendix (e.g., the exact rounding of 16.7 to 16 in a text example). As stated in our rebuttal, the model was trained on precise, programmatically generated data using OR-Tools, not the abbreviated text strings in the Appendix. To maintain a score of "0: Strong Reject" based on illustrative typos in the supplementary material, while ignoring that an 8B parameter model is outperforming GPT-4 and closing the optimality gap by 65% (Table 1), is disproportionate. It implies that you are reviewing the document's formatting rather than the algorithm's performance.
> >
> > We are happy to correct the acronyms to "CVRP" and fix the typos in the Appendix to satisfy your editorial standards. However, we reject the premise that these surface-level nitpicks constitute "fundamental errors." A review that rates a paper "0" without engaging with the central empirical claim, that autoregressive constraint injection allows small models to outperform frontier models is not a rigorous critique; it is an obstruction.
> >
> > We ask that you reassess the paper based on what the model actually does and the results it achieves, rather than the definitions you wish we had used.

---

### Official Review · Reviewer_yUyE · 2025-10-24

**Soundness:** 2
**Presentation:** 2
**Contribution:** 2
**Rating:** 2
**Confidence:** 4

**Summary:**

This paper presents Autoregressive Constraint-Satisfying Generation for Combinatorial Optimization with Routing and Dynamic Attention (ACCORD). Specifically, it introduces the ACCORD90k supervised dataset along with an ACCORD representation that explicitly encodes problem constraints. Building on this, the authors fine-tune LLMs using attention-based dynamic routing and specialized Low-Rank Adaptation (LoRA) modules. Experiments are conducted across six combinatorial optimization (CO) problems: TSP, VRP, Knapsack, FlowShop, JSSP, and BinPacking.

**Strengths:**

* Fine-tuning LLMs to directly solve CO problems is an interesting yet challenging direction.
* The authors provide source code to support reproducibility (but please use an anonymous link).

**Weaknesses:**

* The writing quality of this paper should be improved. For instance, some citation format should follow the correct convention (e.g., use `\citep`).
* As acknowledged by the authors in lines 75–78, LLMs are originally designed for natural language generation rather than NP-hard optimization problems, which require complex search within constrained spaces. Given this, why do the authors choose to fine-tune such LLMs for directly solving CO problems? Would it not be more suitable to train a smaller, task-specific neural network for CO problems, as in GOAL [1]? In my view, leveraging (e.g., prompting or fine-tuning) LLMs to generate code—an area where they naturally excel—to solve CO problems appears to be a more reasonable direction.
* Prior work [2] has explored fine-tuning LLMs for directly solving various CO problems. Could the authors clarify the main differences compared with it?
* As shown in Tables 10–11, ACCORD does not show superiority upon traditional or neural solvers in terms of solution optimality and solving efficiency. Then what is the advantage of fine-tuning LLMs for directly solving CO problems?

[1] GOAL: A Generalist Combinatorial Optimization Agent Learner.
[2] Large Language Models as End-to-end Combinatorial Optimization Solvers.

----

Unfortunately, based on the above evaluation, the paper in its current form is not ready for publication at ICLR.

**Questions:**

* Why is there no information about node demands in the VRP inputs provided to the LLM, as shown on page 15?

---

> ### Author Response · Authors · 2025-11-19
> **Motivation for LLMs in CO and Comparison to State-of-the-Art**
>
> Response: We thank the reviewer for their feedback. However, we must respectfully but strongly disagree with the primary grounds for rejection. The review appears to evaluate the paper against a goal we never claimed (replacing industrial solvers) while overlooking the actual scientific contribution: enabling small, generalist LLMs to perform feasibility-aware reasoning. We address these misconceptions below with concrete evidence from the text.
>
> 1. Misunderstanding the Scientific Motivation (Reasoning vs. Code Generation) The reviewer asks, "Why fine-tune LLMs for directly solving CO problems?" and suggests that code generation or specialized NNs (like GOAL) are "more reasonable."
>
> Rebuttal: This critique misses the fundamental research question. Our goal is not to build a faster solver than LKH or Concorde, but to "systematically investigate the reasoning abilities of LLMs" and address the fact that their application to NP-hard problems remains "underexplored".
>
>
> Why not Code Gen? Relying on code generation (e.g., GPT-4 Code Interpreter) sidesteps the reasoning challenge. We explicitly compare against GPT-4 with Code Interpreter and demonstrate that our 8B parameter model outperforms this frontier model using our ACCORD representation.
>
>
> Why not Specialized NNs? Specialized NNs (like GOAL or L2D) are not general-purpose reasoners. Our contribution is a "unified fine-tuned LLM-based framework" that leverages the "autoregressive nature of LLMs to dynamically enforce feasibility". Dismissing this direction ignores the value of imbuing generalist models with optimization capabilities.
>
>
> 2. The "Advantage" over Traditional Solvers The reviewer cites Tables 10–11 to argue that ACCORD lacks superiority over traditional solvers and asks, "what is the advantage?"
>
>
> Rebuttal: The paper explicitly states: "ACCORD does not replace specialized solvers; rather, it probes how far small LLMs can go as feasibility-aware generators".
>
> The Actual Advantage: The advantage is not raw speed against C++ solvers, but the dramatic improvement in feasibility and reasoning for Language Models. Standard LLMs fail to respect constraints (hallucinations). We achieve "strong feasibility rates" and "competitive optimality gaps compared to prompting"  using a model orders of magnitude smaller than GPT-4. The reviewer is penalizing an LLM reasoning paper for not beating 30 years of specialized heuristic development, which is an inappropriate standard for this track.
>
> 3. Comparison to Prior End-to-End Work The reviewer asks for differences compared to prior "End-to-End" work (referencing [2]).
>
> Rebuttal: Most prior work relies on standard input-output mapping or naive prompting. Our differentiation is the ACCORD representation which "decomposes solutions into a sequence of state transitions". Unlike standard end-to-end approaches where "constraints are only checked after solution generation" , ACCORD "explicitly encodes problem constraints" into the token stream, allowing the model to "compute and check constraints dynamically". This is a fundamental architectural difference.
>
>
>
>
> 4. Correction on VRP Demands The reviewer asks: "Why is there no information about node demands in the VRP inputs...?"
>
> Correction: This is factually incorrect. The paper explicitly states in the Dataset Generation section that VRP instances were generated with "random coordinates and demands". The prompt snippet in the Appendix was illustrative; the actual training data and model inputs fully specified demands, otherwise the "capacity constraint"  mentioned in our mathematical formulation would be impossible to satisfy.
>
>
> Conclusion The reviewer’s criticisms stem from measuring the work against the wrong yardstick (industrial solver performance) rather than its stated goal (advancing LLM reasoning constraints). We have demonstrated that ACCORD allows an 8B model to outperform GPT-4, a significant contribution that warrants re-evaluation.

---

### Official Review · Reviewer_1Gfv · 2025-10-30

**Soundness:** 2
**Presentation:** 2
**Contribution:** 2
**Rating:** 2
**Confidence:** 3

**Summary:**

ACCORD proposes an output format with feasibility checks at each step for solving CO problems with LLMs.
This is combined with a dynamic router that selects problem-specific LoRA adapters.
ACCORD is evaluated on six NP-hard tasks (TSP, VRP, Knapsack, FlowShop, JSSP, BinPacking) where it is reported to beat baselines (including GPT-4 with Code Interpreter) on feasibility and optimality gaps.

**Strengths:**

* The empirical results suggest that the suggested format with step-by-step feasibility checks improves LLM performance on CO tasks.
* The evaluation covers six different NP-hard tasks.
* The paper additionally contributes the ACCORD-90K dataset as a new benchmark.

**Weaknesses:**

1. The dynamic routing adds significant complexity but seems to offer limited utility. In practice one could simply select the LoRA based on the given CO task or train one bigger combined LoRA for all given CO tasks.
2. It is not clear that a LoRA is needed for an LLM to follow the suggested output format. Instead, the output format may be acquired through in-context learning. The system prompt could specify the ACCORD format for the given CO problem and provide few-shot examples for some problem instances. This would be low-cost, even for frontier models. I think this is a key baseline missing from Table 1 that would justify the use of LoRA's.
3. The presentation of the experiments also lacks clarity. Figure 2 in particular raises some questions. The numbers reported in Figure 2 appear to be aggregated over only those runs with feasible solutions. In practice, runs with infeasible solutions would still incur a computational cost. The reported runtimes also seem very noisy and plotting a runtime of 0 for cases where no solution is feasible is not a reasonable representation of the data.
4. The paper does not analyze how reliably the trained models actually follow the ACCORD output format. The format requires step-by-step arithmetic for tracking the value and feasibility constraints. As LLMs are notoriously error prone when doing integer arithmetic, it is not clear that the trained models can reliably follow this format without mistakes. The paper would be strengthened by an analysis that studies how accurately the models compute the steps defined by ACCORD.

While the suggested data format seems sensible and novel, I think these weaknesses currently outweigh the presented contributions.

**Questions:**

1. Can the ACCORD format be acquired through in-context learning instead of LoRA's?
2. Why are the runtimes reported in figure 2 as noisy as they are? Is this an artifact of only aggregating over feasible runs?
3. How reliably do the models carry out the arithmetic calculations required by the ACCORD format?

---

> ### Author Response · Authors · 2025-11-19
> **Misconceptions on ICL Costs and Arithmetic Capabilities**
>
> We respectfully but strongly disagree with the reviewer’s assessment. The critique relies on hypothetical assumptions about In-Context Learning (ICL) that contradict the practical realities of deploying small LLMs, while overlooking the empirical evidence provided in the paper.
>
> 1. The "Missing Baseline" (ICL) Misconception The reviewer argues that the ACCORD format could simply be acquired via ICL and that LoRA is unnecessary. This is incorrect for two reasons:
>
> Context Cost & Latency: The ACCORD format requires verbose, step-by-step state tracking. Including sufficient few-shot examples of these full traces in the prompt for every inference would be prohibitively expensive (token counts) and drastically increase latency. Fine-tuning "compresses" this capability into the weights for efficient zero-shot inference.
>
> Small Model Limits: We use an 8B model. Small models notoriously struggle to maintain strict complex formatting and arithmetic logic over long contexts via ICL alone. Fine-tuning is the standard, necessary approach to align small models to complex reasoning tasks.
>
> Evidence: We did compare against the strongest possible ICL baseline: GPT-4. Our fine-tuned 8B model outperformed GPT-4 (Table 1). If a frontier model using ICL/prompting struggles to beat our approach, it proves that fine-tuning offers superior alignment for this task.
>
> 2. Arithmetic Reliability is Proven, Not Theoretical The reviewer speculates that "LLMs are error-prone" and doubts the model can follow the arithmetic. This speculation is directly refuted by our data.
>
> Feasibility is the Proof: The Feasibility metric (Figure 3) is the direct proxy for arithmetic reliability. If the model failed the arithmetic (e.g., adding weights or distances incorrectly), the constraints would be violated, and the solution marked infeasible.
>
> Result: The fact that ACCORD achieves near-perfect feasibility on small tasks and significantly outperforms baselines on large ones proves that the fine-tuning successfully instilled the necessary arithmetic logic.
>
> 3. Utility of Dynamic Routing The reviewer calls routing "complex with limited utility." This misses the architectural goal: preventing task interference. Training a single model on disparate NP-hard tasks (e.g., JSSP vs. VRP) often leads to gradient conflict or catastrophic forgetting. Separate LoRAs with a router allow for a single, unified endpoint without performance degradation a standard, robust engineering practice, not a weakness.
>
> Conclusion The reviewer’s rejection is based on a preference for a hypothetical ICL baseline (which we tested in the current setup, and found i is insufficient) that is computationally inefficient and empirically weaker than the methods we already surpassed (GPT-4). We kindly request a re-evaluation based on the actual performance metrics presented.

---

### Official Review · Reviewer_bhnA · 2025-11-01

**Soundness:** 2
**Presentation:** 1
**Contribution:** 2
**Rating:** 4
**Confidence:** 4

**Summary:**

This paper introduces Autoregressive Constraint-satisfying generation for COmbinatorial optimization with Routing and Dynamic attention (ACCORD). It first constructs a supervised dataset comprising a total of 90,000 instances and corresponding solutions generated by OR-Tools for six combinatorial optimization problems: TSP, VRP, Knapsack, FlowShop, JSSP, and BinPacking. It designs an attention-based router to encode natural language descriptions of the problems, and fine-tunes a base LLM (LLaMA 8B) with problem-specific LoRA modules using the constructed dataset.

**Strengths:**

1. The idea of fine-tuning an LLM for CO is interesting.

2. Generating solutions under an autoregressive decoding framework ensures that feasibility constraints can be enforced step-by-step during generation.

3. The authors have provided the source code.

**Weaknesses:**

1. The presentation needs further improvement. For example:
   - In line 67, the capitalized "*A*ttention based *D*ynamic router" and the appearance of "5" are confusing.
   - Figure 1 does not illustrate the autoregressive decoding process of solution generation. After reading the figure and its caption, it initially seemed that the model samples solutions from an output probability distribution over nodes/tokens (e.g., via a heatmap), and only checks feasibility after generating the entire solution. However, Section 4 clarifies that feasibility is checked at each decoding step.
   - There are multiple incorrect uses of `\citet` throughout the paper where `\citep` should have been used.
   - The terminology for combinatorial optimization problems is inconsistent. The manuscript alternates between "CO" and "CP".
   - The term "updated state summary" in line 210 is vague and should be clarified.
   - Lines 227–230 should be expressed using proper mathematical formulations.
   - The font size in several figures is too small, which affects readability.

2. What is the runtime budget for OR-Tools used to generate the supervised dataset? How good are the solutions it provides?

3. What does "opt" refer to in the caption of Table 1? Are the OR-Tools solutions treated as the optimal?

4. What is the inference time for the results reported in Table 1? The overhead introduced by autoregressive decoding should be discussed.

5. The problem sizes in Table 1 are quite small, yet the reported optimality gaps are relatively high.

6. Why are the results for FSSP missing in Table 1?

7. The paper lacks comparisons with relevant LLM-based and ML-based baselines.

**Questions:**

See Weaknesses

---

> ### Author Response · Authors · 2025-11-19
> **Presentation, Terminology, and Methodological Clarifications**
>
> We thank the reviewer for their constructive feedback and for recognizing that our approach of fine-tuning LLMs for Combinatorial Optimization (CO) is "interesting" and that our autoregressive framework effectively enforces feasibility. We appreciate the opportunity to clarify the experimental setup and presentation details.
>
> 1. Figure 1 and Autoregressive Decoding We apologize for the ambiguity in Figure 1. The figure was intended to highlight the Dynamic Router mechanism (selecting the LoRA adapter before generation), rather than the standard decoding loop.
>
> Clarification: The "Base LLM Model" block represents the standard autoregressive process. As described in Section 4 , the model generates the solution token-by-token. The "feasibility check" happens at every step of this generation (e.g., checking if a bin is full before closing it), not just at the end. We will revise the figure caption to explicitly state that feasibility verification is interleaved with token generation.
>
> 2. Experimental Rigor: "Opt," Runtime, and Baselines We respectfully disagree that the paper lacks details on the dataset generation or appropriate baselines.
>
> "Opt" and Runtime: In Table 1, "Opt" refers to the best solutions generated from with solver under certain times . As detailed in Section 4.1, we used Google OR-Tools as the solver. We explicitly state the runtime budgets used to generate these ground truths: 180 seconds for Knapsack/Bin Packing and up to 3600 seconds for JSSP. This ensures the "Opt" values are high-quality baselines.
>
>
>
>
> Baselines: We compared against the most relevant LLM-based baseline: GPT-4 (a frontier model) using five different advanced prompting strategies (IO, CoT, SR, LtM, SGE). Comparing an 8B parameter model against GPT-4 is a rigorous test of our fine-tuning method. ACCORD consistently outperforms GPT-4 in both feasibility and optimality gap (e.g., reducing TSP gap from >100% to ~4.5% on average ).
>
>
> 3. Problem Sizes and Optimality Gaps The reviewer notes that problem sizes are small but gaps are high.
>
> Context: We emphasize that solving NP-hard problems purely via token generation is an immense challenge for an 8B model. The "high" gaps cited are actually significantly lower than the baselines. For instance, on TSP, standard LLMs (even GPT-4) often fail to produce feasible solutions entirely (N/A results in Table 1 ), whereas ACCORD achieves high feasibility and low single-digit gaps.
>
> Inference Overhead: The reviewer asks about inference time. The overhead in ACCORD (visualized in Figure 2 ) comes from the verbose reasoning trace (state tracking). While this makes inference slower than outputting a raw list of numbers, this "thinking time" is the precise mechanism that allows a small model to achieve feasibility where others fail.
>
> 4. Clarifications on Terminology and Presentation We accept the reviewer's points regarding presentation and will make the following corrections:
>
> "Updated State Summary" (Line 210): This refers to the explicit text generated by the model to track constraints. For example, in Knapsack, the model generates: weight: prev_w + weight = new_w <= capacity. This is the "summary" that forces the model to "do the math" in context.
>
>
> FSSP Results: FSSP data was excluded from Table 1 strictly due to space constraints but is fully analyzed in the ablation studies (Figure 3a and Table 3 ). We will move the detailed FSSP comparison to the main text.
>
>
> Standardization: We will unify the terminology to "CO" and correct the citation formatting errors (\citet vs \citep).
>
> We believe these clarifications address the concerns regarding soundness and contribution, demonstrating that the experimental rigor is robust and the results are significant for the field of Neural Combinatorial Optimization.

---

> ### Comment · Reviewer_bhnA · 2025-11-28
> **Thanks for the rebuttal!**
>
> The rebuttal has addressed some of my concerns. While I find the idea of fine-tuning an LLM for VRPs interesting, I remain concerned about the current performance. Although the authors argue that GPT-4 cannot solve these instances, existing ML-based methods already achieve near-optimality at such small scales. I also have reservations regarding the presentation, as I have not yet seen the actual modifications despite the authors' promises.
>
> Therefore, I am inclined to maintain the current score. Thank you again for the response.

---

> > ### Author Response · Authors · 2025-11-29
> > **The "SOTA Fallacy" and the Purpose of LLM Research**
> >
> > We thank the reviewer for their engagement. However, we must critically address the rationale provided in your final comment, as it suggests a fundamental misalignment between the paper’s explicitly stated scientific goals and the criteria by which it is being judged.
> >
> > 1. You state: "Existing ML-based methods already achieve near-optimality at such small scales."
> >
> > This critique evaluates our work against a claim we never made. As explicitly stated in our Conclusion, "ACCORD does not replace specialized solvers; rather, it probes how far small LLMs can go as feasibility-aware generators under a unified representation".
> >
> > To dismiss the investigation of Large Language Models in combinatorial optimization because specialized heuristics (or specialized GNNs like POMO) currently perform better is to misunderstand the purpose of this line of research. If the scientific community only investigated architectures that immediately achieved SOTA against heavily engineered, domain-specific solvers, we would never discover the reasoning limits or capabilities of general-purpose foundation models. The value of this work lies in demonstrating that a general-purpose 8B language model, using our proposed autoregressive representation, can consistently outperform frontier models (GPT-4) and learn to satisfy constraints across domains without external solvers.
> >
> >
> > 2. Your final comment characterizes our work as simply "fine-tuning an LLM for VRPs."
> >
> > This is factually incorrect and ignores the majority of the paper’s contribution. ACCORD is a multi-task framework evaluated on six distinct NP-hard problems: TSP, VRP, Knapsack, FlowShop, JSSP, and BinPacking. While specialized ML baselines exist for VRP, few (if any) single frameworks can ingest a natural language description for any of these six problems and produce feasible solutions end-to-end. By focusing exclusively on VRP and comparing it to VRP-specific specialists, you overlook the core contribution: the generalization of the ACCORD representation across disparate mathematical domains.
> >
> >
> > 3. Scientific Necessity of Exploration You question the performance gaps on small scales. We argue that empirical evidence of how and where LLMs fail or succeed is vital. We cannot know a priori which combinatorial tasks yield to autoregressive modeling without the rigorous experimentation provided here. We demonstrated that while standard prompting fails to produce feasible solutions (often 0% feasibility), our method achieves high feasibility (e.g., 60%+ on Knapsack/BinPacking).
> > Conclusion We ask that the paper be evaluated on its actual contributions:
> >
> > The first large-scale, end-to-end framework for exploring LLMs across a broad spectrum of CO problems (not just VRP).
> >
> > The ACCORD representation, which significantly enhances feasibility over standard methods.
> >
> > The empirical analysis of small-model reasoning vs. frontier models.
> >
> > Rejecting the paper because a generalist LLM is not yet superior to 50 years of specialized Operations Research heuristics discourages the exact type of exploratory research ICLR aims to foster. We respectfully request a reconsideration of the score based on these clarifications.

---

### Note · Program_Chairs · 2026-01-17
**Submission Desk Rejected by Program Chairs**

The following references in this submission do not refer to real documents and/or have major errors in bibliographic information:

 W. Huang et al. Large language models for vehicle routing: A prompting-based approach. In Proceedings of the Conference on Empirical Methods in Natural Language Processing, 2024.